# Nucleosome-bound NR5A2 structure reveals pioneer factor mechanism by DNA minor groove anchor competition

Wataru Kobayashi[1], Anna H. Sappler[2], Daniel Bollschweiler[3], Maximilian Kümmecke[1], Jérôme Basquin[4], Eda Nur Arslantas [1], Siwat Ruangroengkulrith [1], Renate Hornberger[1], Karl Duderstadt [2,5] & Kikuë Tachibana [1]✉

Gene expression during natural and induced reprogramming is controlled by pioneer transcription factors that initiate transcription from closed chromatin. Nr5a2 is a key pioneer factor that regulates zygotic genome activation in totipotent embryos, pluripotency in embryonic stem cells and metabolism in adult tissues, but the mechanism of its pioneer activity remains poorly understood. Here, we present a cryo-electron microscopy structure of human NR5A2 bound to a nucleosome. The structure shows that the conserved carboxy-terminal extension (CTE) loop of the NR5A2 DNA-binding domain competes with a DNA minor groove anchor of the nucleosome and releases entry-exit site DNA. Mutational analysis showed that NR5A2 D159 of the CTE is dispensable for DNA binding but required for stable nucleosome association and persistent DNA 'unwrapping'. These findings suggest that NR5A2 belongs to an emerging class of pioneer factors that can use DNA minor groove anchor competition to destabilize nucleosomes and facilitate gene expression during reprogramming.

Gene regulation by transcription factors is important for cell identity. Transcription factors scan chromatin and bind to their cognate motifs in *cis*-regulatory elements, such as promoters and enhancers[1]. DNA access is restricted by nucleosomes, which can inhibit transcription factor binding[2,3]. Transcription factors with pioneer activity, hereafter referred to as pioneer factors, can bind to their (partial) motif on the nucleosome and elicit local opening of closed chromatin[4–6]. The first transcription factor designated as a pioneer factor, FoxA1, replaces the linker histone H1 (refs. 7,8). Recent structural studies of nucleosomes bound by pluripotency-related pioneer factors of the OCT4 and SOX families showed several binding modes and structural changes that depended on the motif position on the nucleosome[9–14]. Binding of other pioneer factors, such as GATA3, has not been found to alter nucleosome architecture[15]. The diverse mechanisms underlying

how pioneer factors engage with nucleosomes therefore remain poorly understood.

Although several pioneer factors interact with the DNA major groove, certain nuclear receptors have dual sequence recognition, simultaneously binding the major and minor grooves[16]. A notable pioneer factor in this class is the orphan nuclear receptor Nr5a2 (also known as liver receptor homolog-1, Lrh-1), which has essential functions in development and disease (we refer to the mouse and human orthologs as Nr5a2 and NR5A2, respectively). Nr5a2 regulates gene expression in adult tissues, including liver and pancreas, and its haploinsufficiency is linked to gastrointestinal and pancreatic cancer[17]. Nr5a2 and Esrrb together are essential for maintaining naive pluripotency in mouse embryonic stem cells[18]. Recently, Nr5a2 has been identified as a key regulator of zygotic genome activation (ZGA) in murine embryos[19]

[1]Department of Totipotency, Max Planck Institute of Biochemistry (MPIB), Munich, Germany. [2]Structure and Dynamics of Molecular Machines, MPIB, Munich, Germany. [3]Cryo-EM Facility, MPIB, Munich, Germany. [4]Department of Structural Cell Biology, Crystallization Facility, MPIB, Munich, Germany. [5]Department of Bioscience, Technical University of Munich, Garching, Germany. ✉e-mail: tachibana@biochem.mpg.de

(*Nr5a2*-knockout embryos have been reported to enter arrest shortly after ZGA[20,21]; however, this could be due to residual maternally deposited Nr5a2 protein in these embryos[19]). Interestingly, Nr5a2 can replace Oct4 to potently improve the efficiency of induced reprogramming of somatic cells to pluripotency[22].

How Nr5a2 engages with nucleosomes to facilitate extreme gene expression changes, such as those during natural and induced reprogramming, is not known. The protein consists of a putative ligand-binding domain (LBD), whose ligand is unknown, and a DNA-binding domain (DBD) that is conserved throughout metazoans[23] (Extended Data Fig. 1a). The DBD consists of two modules of $Cys_4$-type zinc fingers, a carboxy-terminal extension (CTE) and a fushitarazu-factor 1 (Ftz-F1) domain (Extended Data Fig. 1b,c). A crystal structure of the human NR5A2–DNA complex showed that the α-helix of the zinc finger domain and the CTE bind to the major and minor groove, respectively[24] (Extended Data Fig. 1b). Unlike many other nuclear receptors, Nr5a2 binds as a monomer to its DNA motif, TCAAGG(C/T)CA, on naked DNA and nucleosomes[19,24,25]. How orphan nuclear receptors, such as Nr5a2, engage with nucleosomes to exert their pioneer activity is not known.

## Results

### NR5A2 unwraps nucleosomal DNA from histones

To investigate the mechanism underlying NR5A2 pioneer function, we performed selected engagement on nucleosome sequencing (SeEN-seq) to identify the preferred binding site on the nucleosome to generate a homogeneous population of protein complexes for cryogenic electron microscopy (cryo-EM)[10]. The Nr5a2 motif was tiled at 1-base-pair (bp) intervals through a 601-nucleosome positioning sequence, to simultaneously interrogate all potential transcription factor binding registers[26] (Extended Data Fig. 2a–d). Electrophoretic mobility shift assay (EMSA) showed band shifts corresponding to NR5A2 or Nr5a2 with nucleosome libraries (Extended Data Fig. 2e,f). SeEN-seq revealed that both proteins preferentially targeted the entry-exit sites on the nucleosome, as was shown previously using 5-bp intervals[19] (Fig. 1a, Extended Data Fig. 2g,h and Supplementary Table 1). Both orthologs displayed a similar binding pattern with a ~10-bp periodicity (for example, motif positions 10, 20, 30 and 40), reflecting (partial) motif accessibility on the rotational turns of the DNA double helix (Fig. 1a and Extended Data Fig. 2g,h). Because NR5A2 had a stronger binding-site preference than did Nr5a2, we performed all subsequent experiments with the human protein.

To determine which motif position to investigate further, we used modeling to superimpose the DBD onto the canonical nucleosome at the individual binding sites that had been identified by SeEN-seq (Extended Data Fig. 2i). The DBD showed a good fit onto the nucleosome, with no steric hindrances for motif positions at superhelical location (SHL) −6, −5, −4 or −3. By contrast, the DBD showed clashes with histones at SHL +0.5, +2.5, +3 and +5.5 (Fig. 1b and Extended Data Fig. 2i,j). On the basis of this result, we hypothesized that NR5A2 binding to the latter positions would affect nucleosome architecture, and thus could provide insight into NR5A2's pioneer-factor mechanism.

To test this, we selected the strong binding site with the motif positioned at SHL +5.5 for subsequent experiments (Extended Data Fig. 2i). To exclude the possibility that insertion of the Nr5a2 motif into 601 DNA affects DNA end flexibility, we determined the structure of the NR5A2-unbound nucleosome$^{SHL+5.5}$ by cryo-EM and found that it corresponds to the canonical nucleosome structure (Fig. 1c, Extended Data Figs. 3 and 4 and Table 1). Therefore, any structural changes in the NR5A2-bound nucleosome must be due to the binding of NR5A2, not the motif insertion.

The structure of the human NR5A2–nucleosome complex$^{SHL+5.5}$ was determined at a resolution of 2.58 Å (Fig. 1d, Extended Data Figs. 3 and 4 and Table 1). The structure showed an extra density on the nucleosomal DNA that could be fitted well with the crystal structure of the NR5A2 DBD–DNA complex (Extended Data Fig. 4j,k). The other regions, including the LBD, were not visible, presumably because of the flexible nature of the hinge region (Fig. 1d). A prominent feature of the NR5A2–nucleosome complex$^{SHL+5.5}$ is a reorientation of the DNA by ~50° compared with the DNA path on the canonical nucleosome, demonstrating that NR5A2 binding leads to 'unwrapping' of DNA from the histones (Fig. 1e). This finding supports the notion that NR5A2 is a pioneer factor[19].

### The NR5A2 CTE loop competes with minor groove anchors

The nucleosome structure is stabilized by minor groove anchors, which are electrostatic interactions between arginine residues of histones and the DNA minor groove (Fig. 2a)[27,28]. In the canonical nucleosome[27], the minor groove at SHL +5.5 is occupied by the minor groove anchor histone H2A Arg77 (Fig. 2b). By contrast, the structure of the NR5A2–nucleosome complex$^{SHL+5.5}$ shows that the minor groove is occupied by the RGGR motif of the CTE loop (Fig. 2c), suggesting that the minor groove anchor by H2A is outcompeted by NR5A2's RGGR motif. Whether the RGGR–minor groove interaction is required for nucleosome binding is unclear, because NR5A2 also binds to the major groove through its zinc-finger domain. We therefore tested whether the minor groove interaction affects nucleosome binding by generating a mutant NR5A2 protein in which Arg162 (**R**GGR) is replaced with Ala (NR5A2-R162A) (Extended Data Fig. 5a). EMSA of NR5A2-R162A showed slightly reduced DNA- and nucleosome-binding activities compared with those of the wild-type protein (Extended Data Fig. 5b,c). Although we cannot exclude that the R162A mutation decreases nucleosome binding indirectly by lowering the DNA-binding affinity of NR5A2's DBD, these results indicate that RGGR–minor groove interactions contribute to nucleosome binding by NR5A2.

We then investigated how the affinity of the DNA motif recognized by the CTE loop influences nucleosome binding in vivo. The cryo-EM structure of the NR5A2–nucleosome complex$^{SHL+5.5}$ showed that the TCA extension DNA region, located at the 5' end of the TCAAGG(C/T) CA motif, is recognized by the CTE loop. Using published data from CUT&Tag and the assay for transposase-accessible chromatin with sequencing (ATAC-seq)[19], we performed de novo motif searches and identified the Nr5a2 motif in both open and closed chromatin regions in mouse two-cell embryos (Fig. 2d). At the Nr5a2-binding sites on open chromatin, the TCA extension in the motif is slightly permissive (Fig. 2e), which can be potentially explained by constitutively open chromatin sites at a subset of *SINE B1* retrotransposable elements that are bound by Nr5a2. By contrast, the TCA extension in the Nr5a2 consensus motif is highly enriched in Nr5a2-binding sites on closed chromatin (Fig. 2e). This implies that Nr5a2 requires competition for minor groove anchoring to bind nucleosomes on closed chromatin. The lack of chromatin opening at these Nr5a2-bound sites could be due to either the absence of cooperative transcription factor binding or the position of the consensus motif on the nucleosome[1,15]. These data support the notion that the CTE loop is required for nucleosome binding by Nr5a2 in vivo.

We further asked whether NR5A2 binding affects nucleosome architecture. To compare the backbone geometries of all histones, the NR5A2–nucleosome complex$^{SHL+5.5}$ was superimposed on the nucleosome$^{SHL+5.5}$. The register of DNA in the nucleosome$^{SHL+5.5}$ was determined from a high-resolution cryo-EM map region near the dyad axis (Extended Data Fig. 6). The root mean square deviation (r.m.s.d.) values of the corresponding Cα atoms showed no change on the NR5A2-unbound side; however, the backbone geometries of histones on the NR5A2-bound side were shifted (Fig. 2f). This suggests that histones are slightly rearranged owing to NR5A2 binding, which could be due to loss of the minor groove anchor and release of DNA from histones.

### NR5A2 D159 is important for nucleosome binding

We next considered whether NR5A2 binding causes structural changes in core histones. The NR5A2–nucleosome complex showed that the

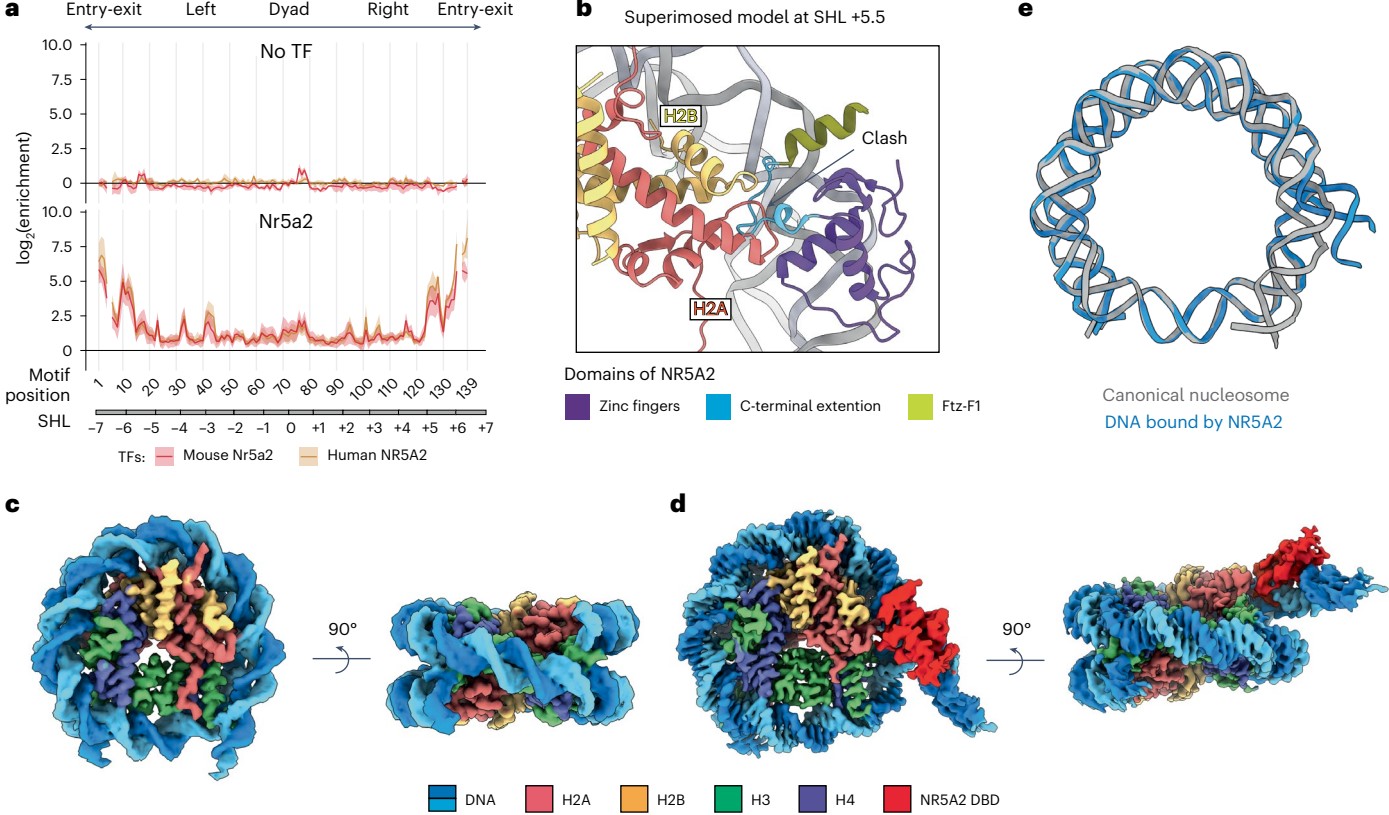

**Fig. 1 | The cryo-EM structure of the human NR5A2–nucleosome complex$^{SHL+5.5}$. a,** SeEN-seq enrichment profiles of mouse Nr5a2 (red) and human NR5A2 (yellow). Enrichment values are plotted against the positions at which the Nr5a2 motif (TCAAGGCCA) starts along the 601 DNA and SHL. Data are shown as mean ± s.d. for three independent experiments. Motif positions 6 and 136 were lost owing to technical issues. **b,** The superimposed model of the canonical nucleosome bound by the NR5A2 DBD[24] (PDB: 2A66) at SHL +5.5.

The CTE loop shows steric clash with histone H2A. **c,** The cryo-EM structure of the 601 nucleosome containing the Nr5a2 motif at SHL +5.5. **d,** The cryo-EM structure of human NR5A2–nucleosome complex$^{SHL+5.5}$ at 2.58Å resolution. **e,** Structural comparison of the DNA path between the canonical nucleosome and the NR5A2–nucleosome complex$^{SHL+5.5}$. The angle of DNA reorientation was measured by Pymol.

side chain of H2A Lys75 (loop 2) was flipped and formed a new hydrogen bond with H2A His82 (Fig. 3a,b). This flip was potentially due to a steric hindrance between H2A Lys75 and NR5A2 Asp159, because these residues collide in a superimposed model of the NR5A2-bound complex on a canonical nucleosome (Fig. 3c). The altered interactions are accompanied by a shift of H2A's α2 helix toward the nucleosome center to accommodate NR5A2 binding (Fig. 3c).

To test whether NR5A2 Asp159 is required for nucleosome binding, we generated NR5A2-D159A (Extended Data Fig. 5d). The D159A mutant bound to naked DNA slightly less well than did wild-type NR5A2, but this difference was not statistically significant (Fig. 3d). By contrast, the D159A mutant showed significantly reduced nucleosome-binding activity compared with that of wild-type NR5A2 (Fig. 3e). We conclude that NR5A2 Asp159 is largely dispensable for naked DNA binding, but is required for nucleosome binding.

### Dynamics of DNA unwrapping by NR5A2

The NR5A2–nucleosome structure shows that NR5A2 binding causes unwrapping of DNA from histones. EMSA cannot be used to assess whether NR5A2 Asp159 promotes the release of DNA (Fig. 3e). To address this, we performed single-molecule Förster resonance energy transfer (smFRET) measurements to observe the real-time dynamics of DNA unwrapping[12,29,30]. The donor fluorophore Cy3 was conjugated to the 5′ end of the 601 DNA, adjacent to Nr5a2 motif at SHL +5.5, and the acceptor fluorophore Alexa Fluor 647 was conjugated to H2A-K119C (Fig. 4a and Extended Data Fig. 7a). Nucleosomes were reconstituted by combining Alexa-Fluor-647-labeled

H2A–H2B histones together with H3–H4 histones, and Cy3-labeled DNA (Extended Data Fig. 7b). Nucleosomes were surface-immobilized for imaging through a biotin–streptavidin interaction (Fig. 4a). Wild-type NR5A2 and NR5A2-D159A showed similar relative binding affinity as the nucleosome without linker DNA, suggesting that extended linker DNA does not affect nucleosome binding (Extended Data Fig. 7c).

Nucleosomes that were not bound by NR5A2 displayed a constant high-FRET state (Fig. 4b), indicating that the DNA was stably wrapped around the histone core on the timescales of the measurements. Upon addition of NR5A2 to nucleosomes, alternation between two FRET states was observed, consistent with DNA unwrapping followed by rewrapping. Multiple transitions between high- and low-FRET states were observed for individual molecules (Fig. 4c,d). The dynamics observed in the presence of NR5A2 were further characterized to determine the kinetics of DNA unwrapping and rewrapping. NR5A2-D159A displayed more frequent switching events, and the duration of its low-FRET (unwrapped) state was, on average, shorter than that of wild-type NR5A2 (wild-type NR5A2: 16.4 ± 1.57 s, NR5A2-D159A: 9.19 ± 0.54 s) (Fig. 4e,g and Extended Data Fig. 7d,e). These results suggest that the unwrapped state of DNA formed with NR5A2-D159A is less stable than is that formed with wild-type protein. By contrast, the average lifetimes of the high-FRET (wrapped) states (wild-type NR5A2: 20.27 ± 2.51 s, NR5A2-D159A: 16.44 ± 2.92 s) are more similar (Fig. 4f,g and Extended Data Fig. 7d,e). Differences in the mean values of the high-FRET periods might be caused by small structural differences in NR5A2 as a result of the substitution. We therefore propose

**Table 1 | Cryo-EM data collection, refinement, and validation statistics**

| | NR5A2–nucleosome complex[SHL+5.5] (EMDB-17740), (PDB 8PKI) | 601 nucleosome[SHL+5.5] (EMDB-17741), (PDB 8PKJ) | 601 nucleosome[SHL+5.5] (EMDB-17741) |
|---|---|---|---|
| **Data collection and processing** | | | |
| Magnification | ×105,000 | ×105,000 | ×22,000 |
| Voltage (kV) | 300 kV | 300 kV | 200 kV |
| Electron exposure (e⁻/Å²) | 65.4 | 65.4 | 65 |
| Defocus range (µm) | −0.6 to 2.2 | −0.6 to 2.2 | −1 to 2.6 |
| Pixel size (Å) | 0.8512 | 0.8512 | 1.885 |
| Symmetry imposed | None | None | $C_2$ |
| Initial particle images (no.) | 3,109,326 | 3,109,326 | 463,334 |
| Final particle images (no.) | 653,440 | 473,492 | 236,337 |
| Map resolution (Å) | | | |
| 0.143 FSC threshold | 2.58 | 2.50 | 4.04 |
| Map resolution range (Å) | 2.29–6.58 | 2.20–5.11 | 4.22–7.61 |
| **Refinement** | | | |
| Initial model used (PDB code) | Ab initio | Ab initio | Ab initio |
| Model resolution (Å) | 2.58 | 2.50 | 4.04 |
| FSC threshold | 0.143 | 0.143 | 0.143 |
| Model resolution range (Å) | Not applicable | Not applicable | Not applicable |
| Map sharpening $B$ factor (Å²) | 78.3 | 83.5 | 199.1 |
| Model composition | | | |
| Non-hydrogen atoms | 11,948 | 11,692 | |
| Protein residues | 816 | 728 | |
| Nucleotides | 275 | 292 | |
| $B$ factors (Å²) | | | |
| Protein | 107 | 36 | |
| Nucleotide | 127 | 69 | |
| R.m.s. deviations | | | |
| Bond lengths (Å) | 0.004 | 0.004 | |
| Bond angles (°) | 0.578 | 0.604 | |
| **Validation** | | | |
| MolProbity score | 1.27 | 1.27 | |
| Clashscore | | | |
| Poor rotamers (%) | 0.62 | 0.84 | |
| Ramachandran plot | | | |
| Favored (%) | 98.49 | 99.02 | |
| Allowed (%) | 1.51 | 0.98 | |
| Disallowed (%) | 0 | 0 | |

that CTE-mediated minor groove competition promotes a stable unwrapped state of the nucleosome.

### Nucleosome binding at the SHL +2.5 position on mouse endogenous nucleosomes

A potential limitation of our experiments is that they are based on a 601 nucleosome-positioning sequence with a motif position near the DNA entry-exit sites. Whether NR5A2 can bind to its motif closer to the dyad axis on an endogenous DNA sequence, and whether Asp159 would be important for such an interaction, is not known. To address these questions, we selected a mouse DNA sequence that is known to be bound by Nr5a2 during ZGA in two-cell embryos[19]. The nucleosome positions in early and late two-cell embryos were identified using published data from micrococcal nuclease digestion with deep sequencing

(MNase-seq)[31]. We identified a genomic locus containing *SINE B1*, at which nucleosome depletion occurs during the transition from the early to late two-cell stage. This suggests that Nr5a2 promotes chromatin accessibility during ZGA in this region (Fig. 5a). To precisely map nucleosome positioning in vitro, we reconstituted a nucleosome with a 180-bp DNA sequence that included *SINE B1*. We found that nucleosomes assembled with Nr5a2 in different positions, and we purified these using a Prep Cell apparatus (Extended Data Fig. 8a–c). Nucleosome fractions were treated with MNase, and MNase-resistant DNA fragments were sequenced by next-generation sequencing (Extended Data Figs. 8d–g and 9). We mapped nucleosomes in which dyad positions were located at 73, 87 or 108 bp, and the Nr5a2 motif was located at SHL +4, +2.5 or +0.5, respectively (Extended Data Figs. 8h and 9). The predominant nucleosome position in vitro contains the Nr5a2 motif

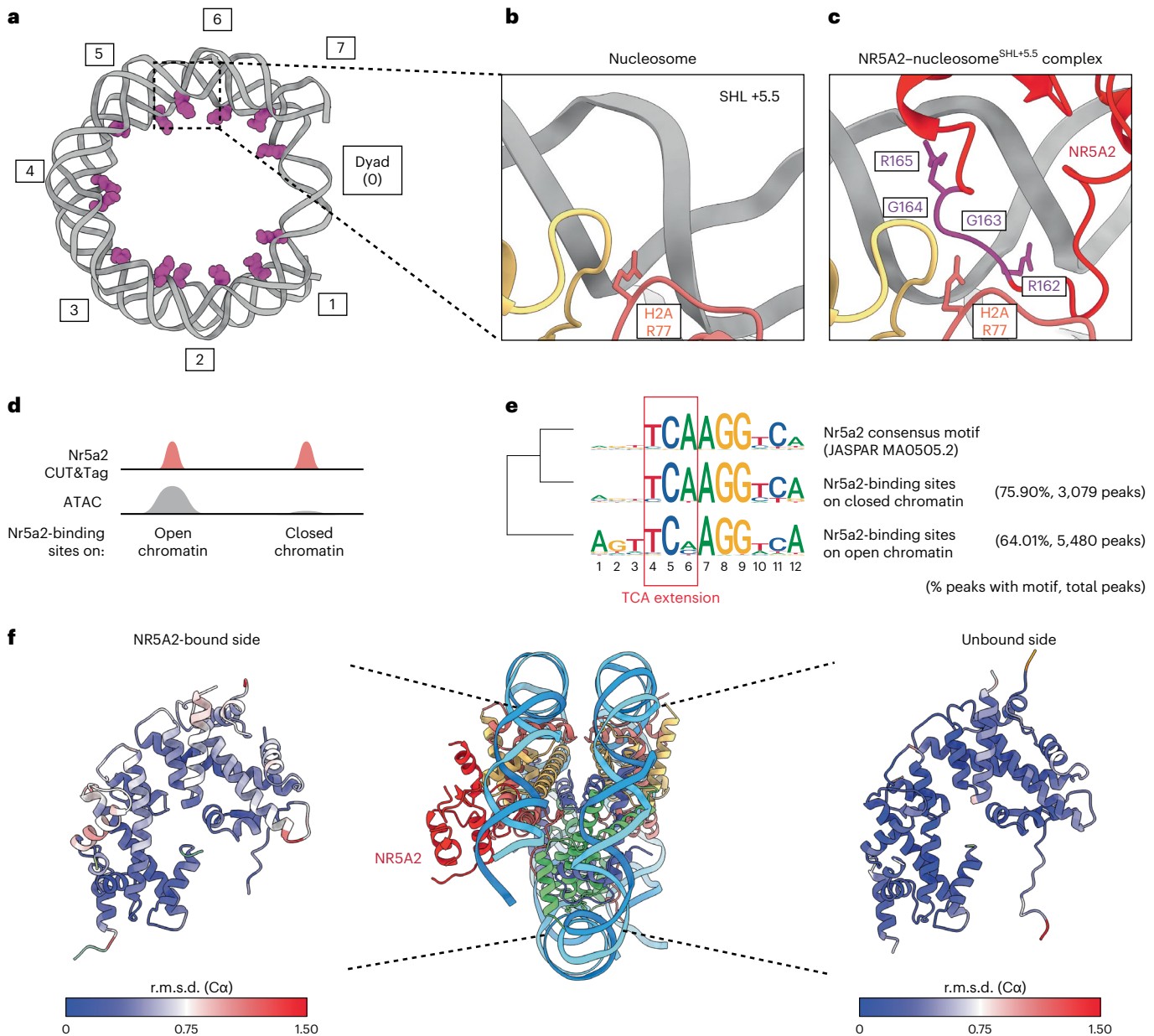

**Fig. 2 | RGGR motif of the CTE loop competes with the minor groove anchor.**
**a**, The nucleosome structure (PDB: 1KX5)[37] showing that the Arg (magenta) residues interacting with the minor groove. Numbers indicate super helical locations. **b**, Close-up view showing minor groove anchor H2A R77 at SHL +5.5 in the unbound structure. **c**, Close-up view showing NR5A2 binding to the minor groove at SHL +5.5 in the bound structure. The RGGR motif (magenta) of the CTE loop occupies SHL +5.5 instead of H2A R77. **d**, An illustration showing Nr5a2 occupancy and chromatin accessibility. **e**, Sequence logos identified each class. The red square shows the TCA extension recognized by the CTE loop. **f**, The r.m.s.d. (Cα) of histones between the unbound and NR5A2-bound nucleosome structure. The r.m.s.d. (Cα) values of histones bound to the DNA gyre at the unbound and NR5A2-bound sites are represented using a blue–white–red color scheme.

at SHL +4 on the left side. In a superimposed model, the Nr5a2 DBD showed no steric clashes with histones at SHL +4, whereas modeling indicated that binding at SHL +0.5 and +2.5 would result in steric clashes with histones (Extended Data Fig. 2i). EMSA showed that NR5A2 binds efficiently to the differently positioned nucleosomes without inducing a shift in register (Extended Data Figs. 8i and 9). We conclude that NR5A2 can bind to multiple positions on nucleosomal DNA derived from a mouse genomic sequence.

We then focused on how NR5A2 overcomes steric clashes with histones at SHL +2.5. We attempted to obtain a cryo-EM structure of this nucleosome bound by NR5A2, but were unsuccessful owing to technical issues. We instead built a superimposed model of the NR5A2 DBD–nucleosome complex[SHL+2.5] (Fig. 5b). The NR5A2 DBD can recognize its own motif on nucleosomal DNA at SHL +2.5 in the vicinity of

histones H3 and H4 (Fig. 5b). The model structure showed that NR5A2 Arg162 in the loop region occupies the minor groove at SHL +2.5, which is bound by minor groove anchor H3 Arg83 (Fig. 5c). This implies that minor groove anchor competition is also relevant for nucleosome binding when the motif is in other superhelical locations. Notably, NR5A2 Asp159 showed a clash with H3 Asp81, suggesting that H3 needs to be accommodated to allow NR5A2 binding (Fig. 5c).

To address whether NR5A2 Asp159 impacts nucleosome binding at SHL +2.5, we purified a 149-bp mouse genomic DNA sequence containing the Nr5a2 motif at approximately SHL +2.5 (Fig. 5d). Wild-type NR5A2 and NR5A2-D159A exhibited similar binding affinities to the endogenous DNA sequence (Fig. 5e). The H2A–H2B complexes used for reconstitution were labeled with Alexa Fluor 647 to detect nucleosome binding (Fig. 5f). Whereas NR5A2 bound to the nucleosome efficiently,

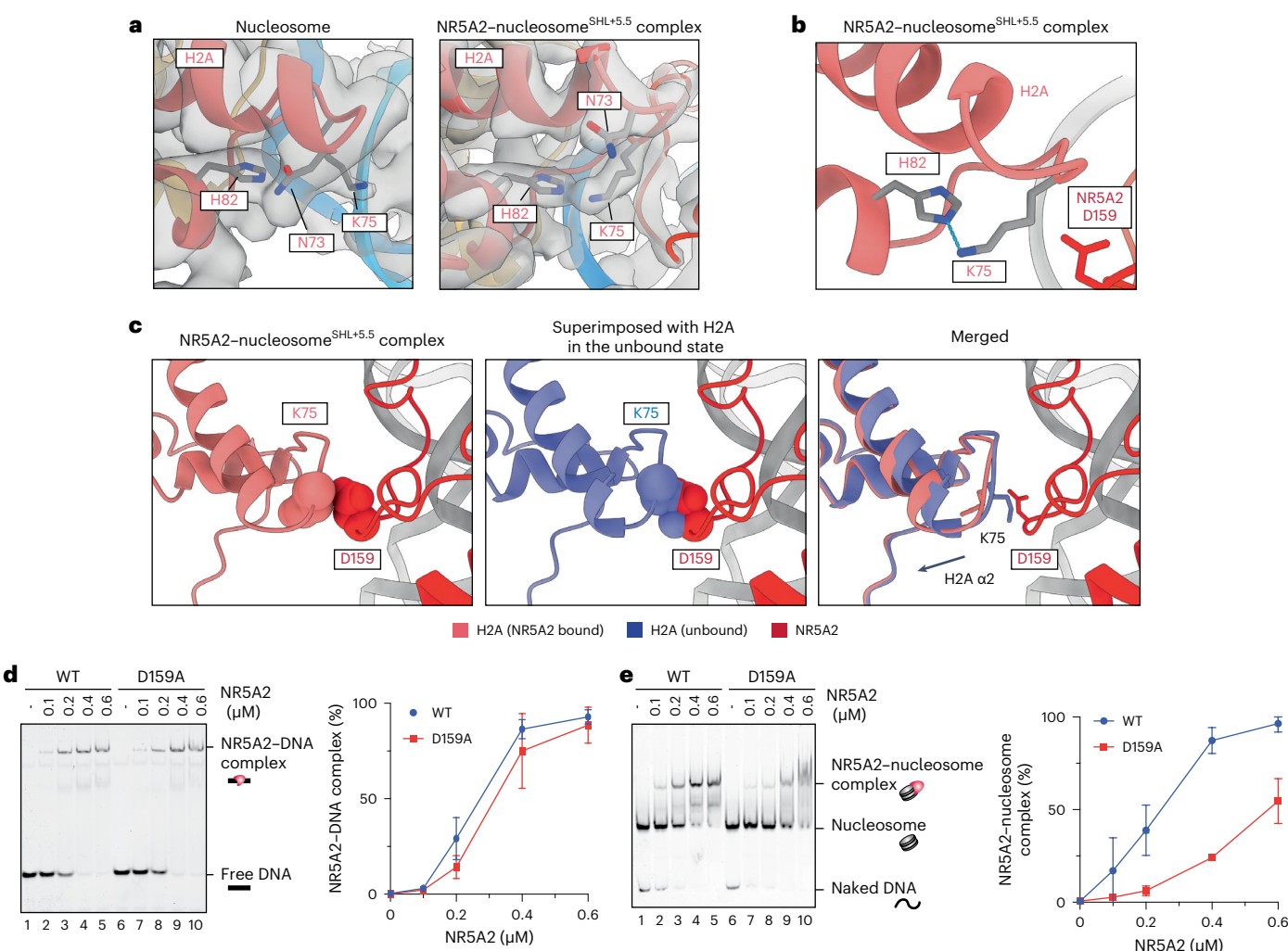

**Fig. 3 | Histone rearrangements by NR5A2. a**, Cryo-EM map density superimposed on a model showing the structural change of histone H2A induced by NR5A2 binding. **b**, Close-up view showing the hydrogen bond between H2A K75 and H2A H82. The blue dotted line covers a distance of 2.97 Å. **c**, Close-up view showing the steric clash between histone H2A and NR5A2. Left and center, the atoms of H2A D159 and H2A K75 are shown as spheres. Right, merged model showing the rearrangement of histone H2A. **d,e**, Left, representative EMSA data of naked DNA (**d**) or nucleosomes containing the Nr5a2 motif at SHL +5.5 (**e**). Right, graphical representations. Data are shown as mean ± s.d. for three independent experiments.

NR5A2-D159A showed significantly decreased binding activity (Fig. 5g). Therefore, NR5A2 Asp159 is also required for binding at SHL +2.5 on a nucleosome reconstituted with an endogenous target sequence. Taken together, our findings suggest that NR5A2 Asp159 contributes to nucleosome binding at several superhelical locations and promotes histone rearrangement.

## Discussion

The structure of the NR5A2–nucleosome complex reveals a previously unknown mechanism of DNA minor groove anchor competition, by which pioneer factors can destabilize histone–DNA interactions. Specifically, the NR5A2-nucleosome structure indicates that the CTE has two functions: (1) replacement of minor groove anchors from histones, and (2) steric clashes that alter the histone arrangement and facilitate stable NR5A2–nucleosome binding (Fig. 6). Although our data do not exclude the possibility that stable binding of NR5A2 to nucleosomes through major groove interaction is required for DNA unwrapping, we propose that DNA release is initiated through minor groove anchor competition and that DNA re-wrapping is prevented by stable binding of NR5A2 to nucleosomes following structural rearrangements in the histone subunits.

The diversity in pioneer factor mechanisms is just beginning to emerge. The OCT4–SOX2 complex causes local DNA distortions and unwraps DNA when bound near the entry-exit sites[10]. OCT4 uses two DNA-binding domains to prevent the re-wrapping of DNA[14]. p53 binds to linker DNA and interacts with the histone H3 tail[32]. Basic helix-loop-helix transcription factors invade between DNA and histones to establish histone interactions[12,13]. Notably, SOX11 binds to its own exposed motif on the nucleosomal DNA, widens the DNA minor groove through DNA bending and pulls DNA away from the histone octamer[9]. Our structure shows that NR5A2 weakens DNA minor groove–histone interactions and directly substitutes these with the RGGR motif in the CTE loop. The exchange in minor groove interactions that occurs in the NR5A2-bound nucleosome provides the first experimental evidence for a theoretical model in which histone–DNA interactions are compensated with pioneer factor–DNA interactions[33]. The loss of histone–minor groove interactions is compensated by the high-affinity binding of NR5A2 to DNA, and results in an energetically stable nucleosome that is invaded by NR5A2. We note that the binding mode depends on the orientation of the motif on the nucleosome. For example, a superimposed model of the DBD on the nucleosome suggests that the CTE loop does not face the histones when the motif occurs in the same orientation on the other DNA gyre (SHL −7 to SHL 0). However, structural and modeling data suggest that the CTE loop can affect minor groove anchors at SHL +2.5 and +5.5 (Figs. 3 and 5). Therefore, motif position and orientation

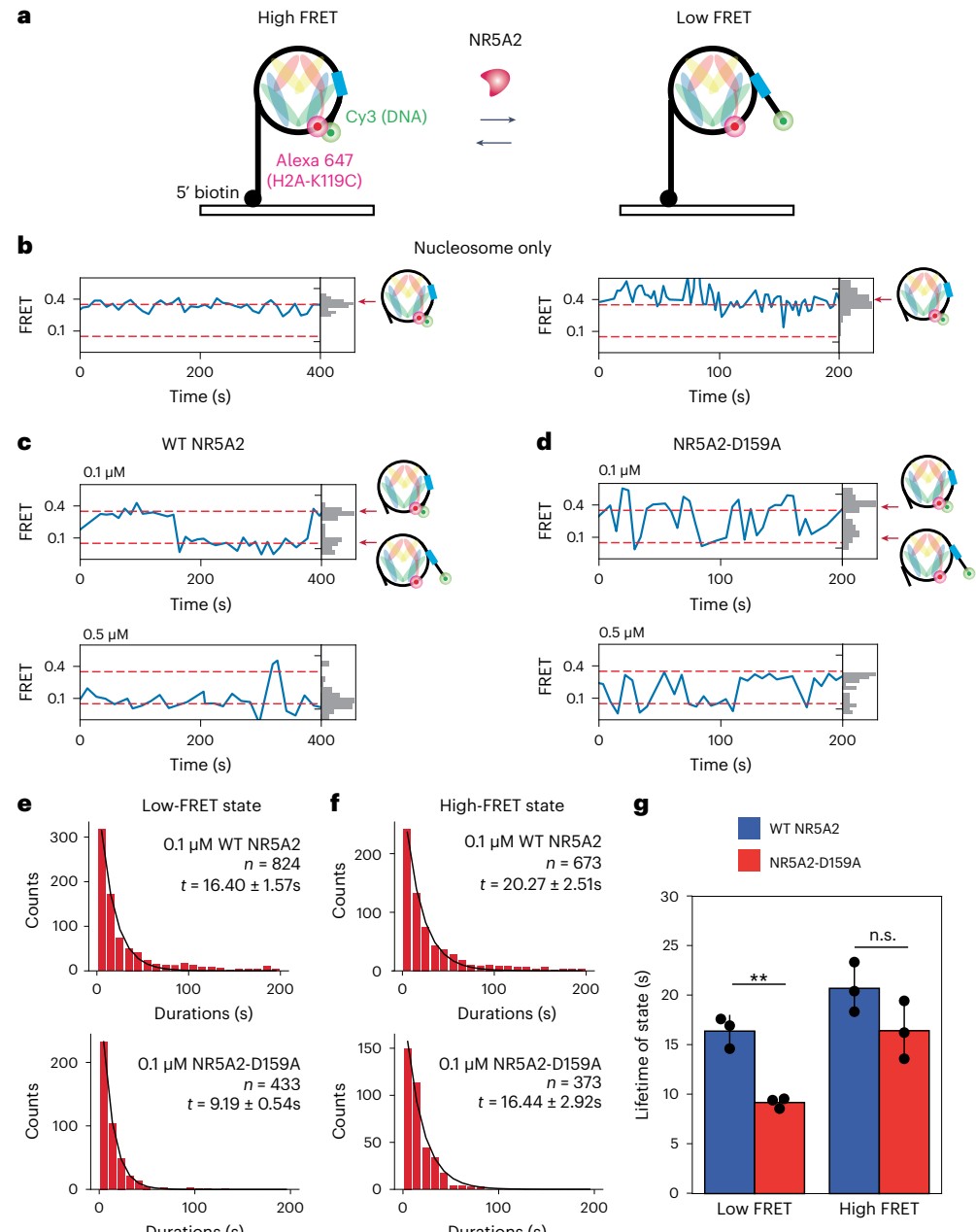

**Fig. 4 | smFRET shows the dynamics of DNA unwrapping and rewrapping.**
**a**, A schematic of smFRET analysis. To monitor the change in FRET efficiency, the donor fluorophore Cy3 was conjugated to the 5′ end of the 601 DNA, adjacent to Nr5a2 motif at SHL +5.5, and the acceptor fluorophore Alexa Fluor 647 was conjugated to histone H2A-K119C. Cyan shows the Nr5a2 motif on nucleosomal DNA. **b**, Representative data of time traces of single nucleosomes. The two dashed lines mark high-FRET and low-FRET states. **c,d**, Representative data of time traces of single nucleosomes in the presence of wild-type (WT) NR5A2 (**c**) and NR5A2-D159A (**d**). Results from experiments with different concentrations of protein (0.1 μM and 0.5 μM) are shown. **e**, Histograms showing the time spent in the low-FRET state, with the number of counts. Results from the wild type (top) and D159A mutant (bottom) are shown. **f**, Histograms showing the duration in the high-FRET state, with the number of counts. Top and bottom, results for the wild type and D159A mutant, respectively. **g**, Quantification of high- and low-FRET states with 0.1 μM wild-type NR5A2 (blue) and NR5A2-D159A (red) for three independent datasets. Statistical analysis was done using an unpaired Student's *t*-test. n.s., not significant.

contribute to the mechanism of DNA minor groove anchor competition by pioneer factors.

A comparison of the preferred motif position of NR5A2 on nucleosomes assembled with the 601 sequence versus a native DNA sequence revealed some differences. SeEN-seq analysis based on 601 nucleosomes with 1-bp tiled intervals of the NR5A2 motif showed that NR5A2 predominantly targets entry-exit sites, including SHL +5.5. On a nucleosome reconstituted with a single endogenous mouse genomic sequence of 180 bp, NR5A2 binds closer to the dyad axis, including

SHL +2.5 and +0.5, as well as to SHL +4. These motif locations also align with peaks in the SeEN-seq data (Fig. 1a). The preferred motif position at SHL +5.5 from 601 SeEN-seq data was not detected on the nucleosome assembled with a native DNA sequence, implying that the binding preferences in the two types of experiments cannot be compared directly. We predict that, if NR5A2 encountered a nucleosome with the motif positioned at SHL +5.5, then this would be the preferred binding site. Differences in motif position on the nucleosome could be at least partly due to genomic DNA being locally loosened compared

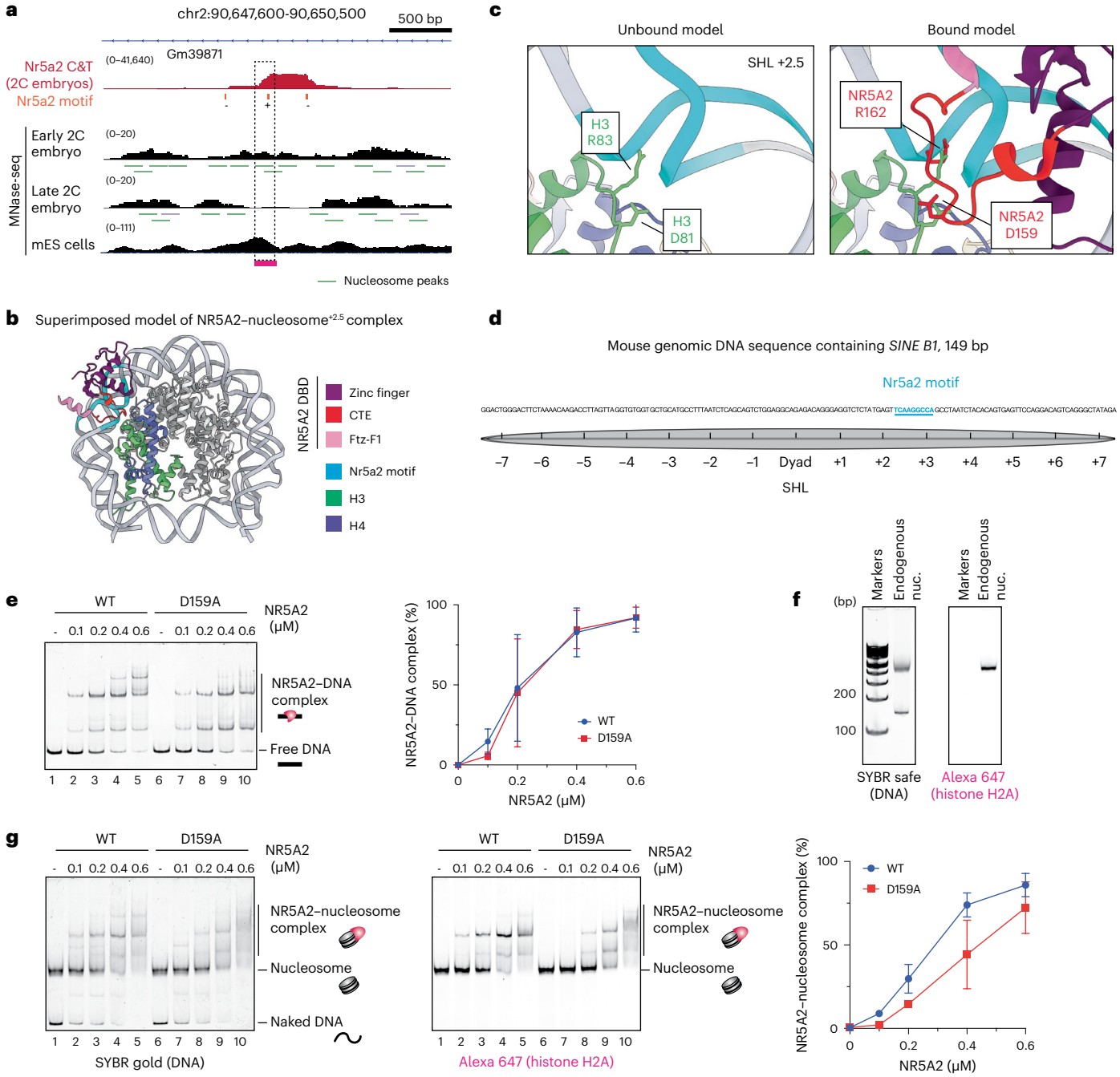

**Fig. 5 | Nucleosome binding with endogenous mouse genomic nucleosome.** **a**, IGV snap snapshot showing Nr5a2 binding overlapped with MNase-seq profile in mouse two-cell (2C) embryos and embryonic stem (mES) cells[31,38]. The dashed rectangles highlight the region used to reconstitute mononucleosomes. **b**, The superimposed model structure of the NR5A2–nucleosome complex[SHL+2.5]. **c**, Left, close-up view showing minor groove anchor H3 R83 at SHL +2.5 in the unbound model structure. Right, close-up view showing NR5A2 binding at the minor groove at SHL +2.5 in the bound model structure. D159 and R162 in the CTE loop show steric clashes with histone H3 D81 and R83, respectively. **d**, DNA sequence of the 149-bp mouse endogenous sequence used for nucleosome reconstitution.

Gray circles show possible nucleosome positions. The Nr5a2 motif located at SHL +2.5 is shown in cyan. **e**, Left, representative EMSA data for endogenous naked DNA. Right, graphical representations. Data are shown as mean ± s.d. for three independent experiments. **f**, Purified 149-bp endogenous mouse genomic nucleosome. Nucleosomes were analyzed by 6% native–PAGE and detected by SYBR safe staining and Alexa Fluor 647 fluorescence. Experiments were performed once. **g**, Left and center, representative EMSA data for the endogenous nucleosome. Alexa Fluor 647 fluorescence signals correspond to the nucleosome. Right, graphical representations. Data are shown as mean ± s.d. for three independent experiments.

with 601 DNA[34,35]. The natural flexibility of DNA may facilitate efficient dyad–proximal nucleosome binding by NR5A2 in vivo.

The zinc finger domain recognizes DNA motifs through a short anchoring α-helix on the major groove, which is a common structural feature associated with pioneer factors[16]. The NR5A2–nucleosome structure shows that the zinc finger domain binds to the major groove

and recognizes the AGGCCA sequence on the nucleosomal DNA. The same superhelical location (SHL +5.5) bound by GATA3 as for NR5A2 shows no effects on nucleosome architecture[15], suggesting that major groove binding is insufficient for DNA release. The RGGR motif in the CTE targets the TCA sequence in the minor groove. This motif is a common feature in members of the NR5A and NR3B orphan nuclear

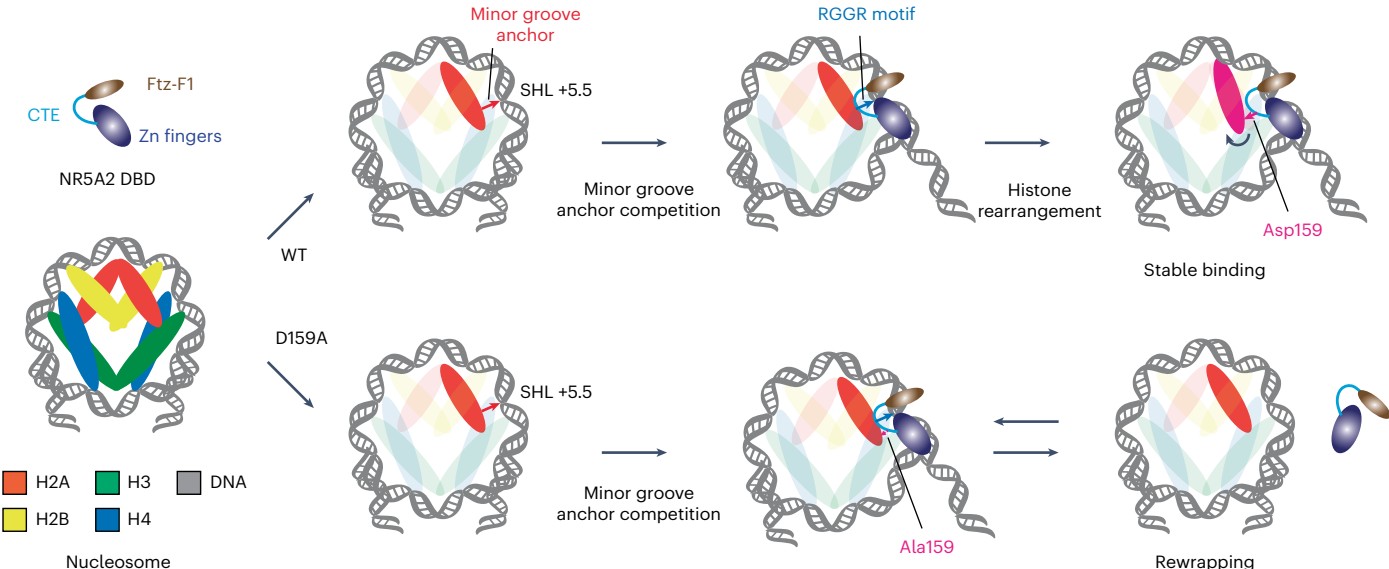

**Fig. 6 | A model of DNA unwrapping by NR5A2.** Zinc fingers and the CTE recognize major and minor grooves in the Nr5a2 motif, respectively. In the wild type, NR5A2 weakens minor groove–histone interaction and substitutes it with the RGGR loop in the CTE. The CTE also induces the histone arrangement to facilitate stable nucleosome binding. Thus, CTE-mediated minor groove competition promotes a stable unwrapped state. By contrast, the CTE in NR5A2-D159A does not stably interact with the minor groove owing to the lack of histone rearrangement. Thus, DNA unwrapping by NR5A2-D159A is less stable than that by the wild type.

receptor families, suggesting that CTE-loop mediated minor groove competition may be conserved among these transcription factors[24] (Extended Data Fig. 1c). Consistent with this notion, the NR3B family member Esrrb targets its own motif on the nucleosome as a pioneer factor[19]. A conserved mechanism of minor groove anchor competition by orphan nuclear receptors could explain their roles in directing cellular potency in natural and induced reprogramming.

The mechanisms by which pioneer factors selectively target cognate motifs and define cell-type-specific regulatory elements remain elusive. A leading model for pioneer factor mechanisms is that multiple pioneer factors cooperatively bind to the nucleosome to outcompete nucleosome assembly[1]. Recent cryo-EM studies have shown how two pioneer factors engage with the nucleosome to induce local destabilization[13,36]. Our structure of the nucleosome bound by NR5A2 at SHL +5.5 suggests that other pioneer factors can access another gyre of DNA (SHL minus position) for cooperative binding. We showed that Nr5a2 preferentially targets its own motif in *SINE B1* during zygotic genome activation[19], but its cooperative binders are still unknown. Future studies of how Nr5a2 cooperates with other transcription factors to target the nucleosome containing *SINE B1* will shed light on the molecular mechanism of ZGA.

## Online content

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

## Methods

### Protein purification

His$_6$-tagged full-length mouse Nr5a2 was purified as described previously[19]. The DNA fragment encoding human NR5A2 was ligated into *NdeI-Bam*HI sites of pET-15b vector (Novagen) with an amino-terminal His$_6$-TEV tag. Human NR5A2 was purified by Ni-NTA agarose (Qiagen) following the same procedure as for mouse Nr5a2 purification. The His$_6$-tag was removed with TEV protease during dialysis, and NR5A2 was further purified by SP sepharose fast flow (Cytiva). The purified sample was dialyzed against buffer 5 (20 mM Tris-HCl (pH 7.5), 400 mM NaCl, 5% glycerol and 1 mM DTT). Human NR5A2-D159A and NR5A2-R162A mutants were purified in the same way as the wild-type protein.

Mouse histones H2A, H2A K119C, H2B, H3.3 and H4 were expressed and purified as previously described[39]. Histone octamer was reconstituted as described previously[39]. H2A-K119C–H2B and H3.3–H4 were reconstituted as previously described. For the fluorescent labeling of the H2A–H2B complex, 50 μM of H2A-K119C–H2B was incubated with 500 μM Alexa Fluor 647 C$_2$ maleimide (Invitrogen) in 20 mM Tris-HCl (pH 7.5) buffer containing 0.1 M NaCl and 1 mM TCEP at room temperature for 2 h in the dark. The reaction was stopped by the addition of 150 mM of 2-mercaptoethanol, and the sample was then dialyzed against 20 mM Tris-HCl buffer containing 0.1 M NaCl, 1 mM EDTA and 5 mM 2-mercaptoethanol.

### SeEN-seq assay

SeEN-seq was performed as previously described[10,19]. Nr5a2 motif (JASPAR ID: MA0505.1, TCAAGGCCA) was tiled at 1-bp intervals across the 147-bp Widom 601 DNA sequence. The pooled DNA library (153 bp) was digested by *Eco*RV and was further purified using a gel-extraction kit. For preparation of the nucleosome library, purified library DNA (3 μg) was mixed with 97 μg spike-in 601 DNA (153 bp). The nucleosome library was reconstituted using a salt dialysis method and was further purified by polyacrylamide gel (6%) electrophoresis using a Prep Cell apparatus (Bio-Rad). The nucleosomes (0.1 μM) were incubated with mouse Nr5a2 (0, 1 or 1.2 μM) or human NR5A2 (0, 0.4 or 0.8 μM) at room temperature for 30 min in a reaction buffer (20 mM Tris-HCl (pH 7.5), 120 mM NaCl, 1 mM MgCl$_2$, 10 μM ZnCl$_2$, 1 mM DTT, 100 μg ml$^{-1}$ BSA). Three independent experiments were performed, and the bands were visualized by SYBR Gold staining (Invitrogen). DNA libraries were prepared as described previously. Purified DNA libraries were sequenced on a NextSeq 500 with paired-end 150-bp reads (Illumina). The DNA sequences are shown in Supplementary Table 1.

### SeEN-seq data processing

SeEN-seq data analysis was performed as described previously[10,19]. In brief, raw reads were mapped to the reference constructs used in SeEN-seq assay by Bowtie2 (ref. 40) (version 2.3.5.1) with the following parameters: -t -q –very-sensitive –no-discordant –no-mixed. Only reads with a map quality of more than 20 were used in further analysis. The SAMtools[41] idxstat command was used to count the number of reads that map to each construct of the SeEN-seq library. The read counts were normalized by their respective library size, before being further divided by the library-size-normalized value of counts of the Widom 601 template construct in their library. The transcription-factor-binding enrichment score of each construct was represented as a log$_2$-transformed fold change between the normalized bound and unbound fraction of each construct. The raw read numbers are shown in Supplementary Table 1.

### Preparation of the nucleosomes

601 DNA containing the Nr5a2 motif at SHL +5.5 was purified with a large-scale plasmid purification using a plasmid containing 16 copies cloned into pGEM-T easy vector (Promega). The endogenous DNA fragments for nucleosome reconstitution were amplified by PCR and further purified by polyacrylamide gel (6%) electrophoresis using a Prep Cell apparatus (Bio-Rad). The eluted DNA was concentrated with an Amicon Ultra centrifugal filter unit (Millipore). The DNA sequences are shown in Supplementary Table 2.

The nucleosomes were reconstituted using a salt dialysis method[39]. For the nucleosome reconstitution with 601 DNA, the DNA and mouse histone octamer were mixed in a 1:1.8–2.0 molar ratio in 2 M KCl. For the nucleosome reconstitution with the endogenous DNA sequence, the DNA, non-labeled or Alexa-Fluor-647-labeled H2A–H2B complex, and H3.3–H4 complex were mixed in a 1:4:3.6 molar ratio in 2 M KCl high-salt buffer. The reconstituted nucleosomes were further purified by polyacrylamide gel (6%) electrophoresis using a Prep Cell apparatus (Bio-Rad). The nucleosomes were concentrated with an Amicon Ultra centrifugal filter unit (Millipore) and were stored in TCS buffer (20 mM Tris-HCl (pH 7.5) and 1 mM DTT) at 4 °C.

### Preparation of NR5A2–nucleosome complex for cryo-EM analysis

The nucleosome (2 μM) in which the Nr5a2 motif was located 128 bp from the 5′ end of DNA (54 bp from the dyad, SHL +5.5) was mixed with human NR5A2 (4 μM) in a reaction buffer (20 mM Tris-HCl (pH 7.5), 120 mM NaCl, 1 mM MgCl$_2$, 10 μM ZnCl$_2$, 1 mM DTT). After incubation at 25 °C in the heat block for 30 min, the sample was purified and stabilized by the Grafix method[42]. A gradient was formed with buffer 1 (10 mM HEPES-NaOH (pH 7.5), 20 mM NaCl, 1 μM ZnCl$_2$, 1 mM DTT and 5% sucrose) and buffer 2 (10 mM HEPES-NaOH (pH 7.5), 20 mM NaCl, 1 μM ZnCl$_2$, 1 mM DTT, 20% sucrose and 2% formaldehyde), using a Gradient Master (BioComp). The reconstituted NR5A2–nucleosome complex$^{SHL+5.5}$ (150 μl × 2) was placed on the top of the gradient solution and was ultracentrifuged at 27,000 r.p.m. (124,668$g$) at 4 °C for 16 h, using an SW41 Ti rotor (Beckman Coulter). The fractions were analyzed by 5% non-denaturing PAGE. The collected samples were then desalted with a PD-10 column (GE Healthcare) equilibrated with 10 mM HEPES-NaOH (pH 7.5) buffer containing 1 μM ZnCl$_2$ and 1 mM DTT and concentrated with an Amicon Ultra centrifugal filter unit (Millipore). The sample of nucleosomes without NR5A2 was purified by the same method.

### Cryo-EM specimen preparation and data acquisition

To prepare the cryo-EM specimen, the sample (4 μl) was applied to a glow-discharged holey carbon grid (Quantifoil R1.2/1.3 200-mesh Cu). The grids were blotted for 3.0 s at a blotting strength setting of 5 under 100% humidity at 4 °C and then plunged into liquid ethane and cooled by liquid nitrogen using a Vitrobot Mark IV (Thermo Fisher). Data acquisition for the NR5A2–nucleosome complex$^{SHL+5.5}$ was conducted using a 300 kV Titan Krios G2 (Thermo Fisher Scientific) equipped with a GIF Quantum 967 energy filter and a K3 direct electron detector (Gatan) running in correlated double sampling mode at a magnification factor of 105kx, equivalent to a pixel size of 0.85 Å per pixel. Automated data acquisition was performed using SerialEM software. A total of 12,510 videos were collected, with a total electron dose of 65.4 e/Å$^2$ fractionated over 40 frames. The data set for pure nucleosomes was acquired on a 200 kV Glacios (Thermo Fisher Scientific) equipped with a K2 Summit direct electron detector (Gatan) at a magnification factor of 22kx, equivalent to a pixel size of 1.89 Å per pixel. Here, 1,998 videos were collected, with a total electron dose of 61 e/Å$^2$ fractionated over 40 frames.

### Image processing

Data processing was performed by CryoSPARC[43]. For the dataset obtained by Gracios, data processing was performed on the fly using CryoSPARC Live (version 4.1) on its default processing pipeline, applying $C_2$ point group symmetry for the final homogeneous refinement. This resulted in a three-dimensional Coulomb potential map at a nominal resolution of 4.04 Å. This map was subsequently post-processed through local *B*-factor correction by DeepEMhancer[44].

## Model building and structural analysis

We have placed a nucleosome model by molecular replacement using the PDBID 3LZ0 ref. [45] as template. In the case of the NR5A2-nucleosome we placed by rigid body fitting in the EM map the NR5A2 model using the PDBID 2A66. The models for the free nucleosome and NR5A2-nucleosome complex were completed by manual rebuilding in COOT[46] and refined using in Phenix real space refinement[47]. The structures were analyzed using ChimeraX[48]. Molecular models were superimposed with chain A using the matchmaker option. The RMSD calculation for each residue was calculated using ChimeraX.

## Electrophoretic mobility shift assay

For EMSA with short oligonucleotide DNA, 25 bp double-stranded DNA (50 nM) containing the Nr5a2 consensus motif (GAGAGAGTCAAGGCCATGGCTCACT) was incubated with NR5A2 at room temperature for 30 min in a reaction buffer (20 mM Tris-HCl (pH7.5), 120 mM NaCl, 1 mM MgCl$_2$, 10 μM ZnCl$_2$, 1 mM DTT, 100 μg ml$^{-1}$ BSA). After the incubation, the samples were loaded onto 10% non-denaturing polyacrylamide gels (0.5×TBE), and electrophoresis was performed at 100 V for 1 h at room temperature. The gels were stained by SYBR Gold (Invitrogen) and were imaged using GelDoc Go imaging system (Bio-Rad).

For EMSA with nucleosomes, nucleosomes (50 nM) were incubated with NR5A2 at room temperature for 30 min in a reaction buffer (20 mM Tris-HCl (pH7.5), 120 mM NaCl, 1 mM MgCl$_2$, 10 μM ZnCl$_2$, 1 mM DTT, 100 μg ml$^{-1}$ BSA). After the incubation, the samples were loaded onto 10% non-denaturing polyacrylamide gels (0.5×TBE), and electrophoresis was performed at 100 V for 70 min at room temperature. The gels were imaged by detecting Alexa Fluor 647 fluorescence and SYBR Gold (Invitrogen) using the ChemiDoc MP imaging system or the GelDoc Go imaging system (Bio-Rad).

For quantification, data from at least three replicates were analyzed using ImageLab (Bio-Rad) and plotted in Prism (GraphPad). Data are shown as mean and s.d.

## MNase assay

Mouse endogenous nucleosomes (180 bp, 50 nM) were incubated at 25 °C for 5 min with the indicated units of MNase (New England Biolabs, M0247S) in 11 μl of reaction buffer (45 mM Tris-HCl (pH7.5), 120 mM NaCl, 1 mM MgCl$_2$, 2.5 mM CaCl$_2$, 10 μM ZnCl$_2$, 1 mM DTT, 100 μg ml$^{-1}$ BSA). The reaction was stopped by the addition of 50 μl of deproteinization solution (20 mM Tris-HCl (pH 8.0), 20 mM EDTA, 0.5% SDS, 0.5 mg ml$^{-1}$ Proteinase K). The resulting DNA was extracted by phenol–chloroform and purified by ethanol precipitation. The samples were loaded onto 10% non-denaturing polyacrylamide gels (0.5×TBE), and electrophoresis was performed at 200 V for 35 min at room temperature.

## DNA sequencing of the nucleosome positioning

Mouse endogenous nucleosomes (180 bp, 50 nM) were incubated at 25 °C for 30 min in the absence or presence of NR5A2. MNase (10 U) was added and further incubated at 25 °C for 5 min in a 11 μl of reaction buffer (45 mM Tris-HCl (pH7.5), 120 mM NaCl, 1 mM MgCl$_2$, 2.5 mM CaCl$_2$, 10 μM ZnCl$_2$, 1 mM DTT, 100 μg/ml BSA). The resulting DNA was extracted and purified in the same manner as for the MNase assay. The samples were separated by non-denaturing polyacrylamide gel (10%) electrophoresis. The bands were visualized by SYBR Gold staining (Invitrogen) using GelDoc Go imaging system (Bio-Rad). To purify MNase-treated DNA fragments, the range from 100 to 200 bp was excised from a polyacrylamide gel. Gel slices were transferred into DNA low-binding tubes and incubated with acrylamide gel extraction buffer (100 μl, 500 mM ammonium acetate, 10 mM magnesium acetate, 1 mM EDTA, 0.1% SDS) and heated to 50 °C for 30 min. After incubation, H$_2$O (50 μl) and QIAquick Gel Extraction kit QG buffer (450 μl, Qiagen) were added, and the samples were further heated at 50 °C overnight. The supernatant containing DNA fragments was purified with QIAquick Gel

Extraction spin columns. DNA libraries were prepared with NEBNext Ultra II libraries Prep Kit for Illumina (E7645) and were sequenced on a NextSeq 500 with paired-end 150-bp reads (Illumina).

## Motif enrichment analysis

To identify the motif of Nr5a2 in different chromatin contexts, the Nr5a2 CUT&Tag and ATAC-seq peaks in two-cell embryos were downloaded from Gassler et al.[19]. DNA motif enrichment for each group of genomic regions was performed using findMotifsGenome.pl command from HOMER[49]. Similarity tree of the de novo discovered motifs was made in R using the ggseqlogo, universalmotif, ape, and ggtree packages[50–54].

## MNase data analysis (in vivo)

Low-quality reads were trimmed and removed by Trim Galore version 0.6.6 (https://github.com/FelixKrueger/TrimGalore) with the option: –paired–quality 20–length 20. The additional options –clip_R1 4–clip_R2 4 were used to trim the two-cell-embryo dataset. Trimmed reads were mapped to mouse genome (mm10) using bowtie2 (version 2.4.4) with -t -q -N1 -L 25 -X 1000–no-discordant–no-mixed options[40]. Unmapped reads and reads with a mapping quality of less than 30 were removed using SAMtools (version 1.14)[41]. PCR duplicated reads were detected and removed using Picard (version 2.26.4). Genome coverage was calculated using the bamCoverage command from deepTools package[55]. For the two-cell-embryo dataset, positions of nucleosomes were identified using nucleR.R scripts from Nucleosome Dynamic CLI[56] (https://github.com/nucleosome-dynamics/nucleosome_dynamics) with additional options: –type paired–minoverlap 20.

## MNase data analysis (in vitro)

Low-quality reads were trimmed and removed using Trim Galore with –paired–quality 20–length 20 option. Then, trimmed reads were mapped to reference reads using bowtie2 (version 2.4.4) with the following option -t -q -N1 -L 25 -X 1000–no-discordant–no-mixed. Reads with a map quality of less than 30 and unmapped reads were filtered out from subsequence analysis. Only reads with fragment length between 145 and 147 bp were kept for further analysis. The middle point of each read fragment was considered a dyad of the nucleosome. The enrichment of dyads at a given position on the reference DNA construct was calculated as a percentage relative to the total number of dyads (referred to as dyad density). The $\log_2$(fold change) of dyad density between the MNase profiles with and without Nr5a2 was used to show the change in dyad positioning in the presence of Nr5a2. Visualization of the dyad enrichment was done in R using the ggplot2 package.

## Single-molecule FRET assay

**PEG–biotin microscopy slide preparation.** Glass coverslips for total internal reflection fluorescence microscopy (TIRF) were prepared as described previously[57]. In brief, 22 mm × 22 mm cover slips (Marienfeld) were cleaned, silanized with 3-aminopropyltriethoxysilane in acetone and incubated with a PEG-/PEG–biotin solution (0.4% (wt/vol) biotin–PEG-succinimidyl carbonate (molecular weight 5,000) and 15% (wt/vol) mPEG–succinimidyl carbonate (molecular weight 5,000) in fresh 0.1 M NaHCO$_3$) overnight. Afterwards, they were washed with ddH$_2$O, dried under an air stream and stored under vacuum.

**Flow cell preparation.** For flow cell assembly, a glass coverslip was incubated with 0.2 mg ml$^{-1}$ streptavidin in blocking buffer (50 mM Tris, pH 7.6, 50 mM KCl, 2% (vol/vol) Tween-20) for 20 min. It was then rinsed with ddH$_2$O and dried under an air stream. A double-sided tape (-0.1 mm thick) with two flow lanes of approximately 0.5 mm width was taped to a glass slide containing entry and exit holds for buffer tubes, and then to the functionalized cover slip forming two flow lanes. Polyethylene tubes with a length of 25 cm (0.58 mm inner diameter) were stuck into the entry and exit holds of the flow lanes and secured in place using epoxy glue. The flow cell was further glued into a custom-built

metal holder to prohibit air diffusion to the sample. The holder was then mounted onto a home-built TIRF microscope. The flow lane surface was coated by flowing blocking buffer at a rate of 100 µl min$^{-1}$ for 5 min, followed by incubation for at least 15 min. It was then washed with working buffer (50 mM Tris pH 7.6, 120 mM NaCl, 1 mM MgCl$_2$, 10 µM ZnCl$_2$, 10% glycerol, 1 mM DTT, 100 µg ml$^{-1}$ BSA) at 100 µl min$^{-1}$ for 5 min.

**Sample preparation and application.** All buffers were degassed prior to use at 40 mbar for 40 min. Nucleosomes were diluted in working buffer (50 mM Tris pH 7.6, 120 mM NaCl, 1 mM MgCl$_2$, 10 µM ZnCl$_2$, 10% glycerol, 1 mM DTT, 100 µg ml$^{-1}$ BSA) to a concentration of about 800 fM. They were applied to the flow lane at a rate of 150 µl min$^{-1}$ for 30 s, incubated on the slide for 20 s and washed out extensively at 150 µl min$^{-1}$ for 5 min. A second wash with imaging buffer (working buffer with 1 mM Trolox (aged for 5 min under ultraviolet light), 2.5 mM PCA, 0.21 U ml$^{-1}$ PCD) was performed (1 min, 150 µl min$^{-1}$). NR5A2 or NR5A2-D159A was applied at different concentrations (either 0.1 or 0.5 µM) in imaging buffer for 2 to 3 min at 150 µl min$^{-1}$. Then, the flow was stopped and the tubes were clipped to avoid oxygen diffusion. Data acquisition was initiated immediately.

**Imaging conditions.** A RM21-micromirror TIRF microscope from Mad City Labs, modified according to Larson et al.[58] and equipped with an Apo N TIRF ×60 oil immersion objective (numerical aperture 1.49, Olympus), was used. The room temperature was controlled at 22.5 ± 0.5 °C. Green laser pulses (OBIS 532 nm LS 120 mW, Coherent) of 200 ms duration and one-fifth of the maximal power were used to excite Cy3 dyes on the nucleosomes.

Scattered light was removed, and emission was separated with emission filter sets (ET520/40m and ZET532/640m, Chroma) and was split at 635 nm using an OptoSplit II dualview (Cairn Research). It was then detected using an iXon Ultra 888 EMCCD camera (Andor). Image-acquisition frequency ranged from 6 s for NR5A2-D159A to 10 s for NR5A2. Autofocusing was performed every 8 to 10 frames while videos of up to 30 min were collected. A total of up to 1 h of measurements was collected for each sample before nucleosome stability decreased.

**Data analysis.** FRET data were analyzed as reported[59], using the Fiji plugin Mars (Molecule Archive Suite). All images were corrected for the Gaussian beam profile of the excitation light source. Populations of nucleosomes containing only a Cy3 dye (donor-only) and nucleosomes with both dyes, displaying FRET, were selected by finding spots in the red detector region and converting them to the green region using an Affine2D matrix. Single spots were integrated, the immediate background was subtracted and the fluorescence over time was analyzed separately for each molecule. Bleaching positions for both donor and acceptor fluorophores were determined by finding the largest change point within the signal. FRET traces were selected on the basis of several criteria: accepted events could display only one bleaching event per fluorophore, show a FRET signal for several frames before bleaching and demonstrate donor fluorescence recovery after bleaching of the acceptor or vice versa. To investigate the dwell times of NR5A2 on nucleosomes, only FRET traces with switching events between high- and low-FRET states were then chosen for further analysis. These events are indicated by a sudden increase or decrease in donor fluorescence and an anticorrelated change in FRET signal.

The FRET efficiency was calculated for each molecule separately after correcting the signal for leakage from green to red channel ($\alpha$), and the differences in detection efficiencies of the two channels as well as quantum yields of the two dyes ($\gamma$) were determined as previously reported[60].

To determine residence times of NR5A2 on the nucleosomes, changes in FRET efficiency over time were investigated. A threshold of 0.48 was chosen to separate low and high efficiency values, corresponding to NR5A2 dwell times on nucleosomes and times with wrapped, unbound nucleosomes. The lifetimes of the states were fitted using single exponential-decay curves for three independent replicates. Binding and unbinding rates were calculated by inverting the half-lives of the respective states for each replicate. An unpaired Student's $t$-test was performed for statistical analysis.

**Reporting summary**
Further information on research design is available in the Nature Portfolio Reporting Summary linked to this article.

## Data availability
The maps are available in the Electron Microscopy Data Bank (EMDB) database under accession no. EMD-17740 (NR5A2-nucleosome complex SHL +5.5) and EMD-17741 (601 nucleosome containing Nr5a2 motif at SHL +5.5). The atomic models are available in the Protein Data Bank database: PDB 8PKI (NR5A2–nucleosome complex SHL +5.5) and PDB 8PKJ (601 nucleosome containing Nr5a2 motif at SHL +5.5). PDB: 3LZ0 ref. 45 and 2A66 were used for model building. PDB: 1KX5 ref. 37 was used as the canonical nucleosome structure. The Nr5a2 CUT&Tag and ATAC-seq datasets were obtained from GSE178234 (ref. 19). The MNase data set for two-cell embryos was downloaded from the Genome Sequence Archive (GSA) accession CRA005944 (ref. 31). The MNase data set for mES cells was downloaded from the Gene Expression Omnibus (GEO) accession GSE82127 (ref. 38). Requests for plasmids generated in this study should be directed to the corresponding author. Source data are provided with this paper.

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

## Acknowledgements

We are very grateful to K. Abe, L. G. Hernandez, C. Kobayashi, K. Straßer and M. Zaczek for their contributions and technical support. We thank N. Thomä for advice on SeEN-seq. We are grateful to A. Musacchio for insightful discussions. We thank J.-M. Peters for critical reading of the manuscript and all members of K.T.'s laboratory for discussions. We thank T. Schäfer at the cryo-EM facility for assistance in cryo-EM data collection, and R. H. Kim for sequencing at the NGS facility, MPIB. K.T. is an Honorary Professor at the Department of Biology, Ludwig-Maximilians-University, Munich. Funding: European Research Council grant ERC-CoG-818556 TotipotentZygotChrom (K.T.). European Research Council grant ERC-StG-804098 ReplisomeBypass (K.D.). Max Planck Society (K.T., K.D.).

## Author contributions

W.K. and R.H. performed protein purification and sample preparation for cryo-EM analysis. M.K. performed SeEN-seq. S.R. analyzed SeEN-seq and MNase-seq data. W.K., D.B. and J.B. performed cryo-EM analysis and model building. W.K. and E.N.A. performed biochemical experiments. A.H.S. and K.D. performed smFRET analyses. K.T. conceived the project and supervised the work. W.K., A.H.S., K.D. and K.T. planned the project, designed the experiments and wrote the manuscript. All authors discussed the results and commented on the manuscript.

## Funding

## Competing interests

The authors declare that they have no competing interests.

## Additional information

**Extended data** is available for this paper at https://doi.org/10.1038/s41594-024-01239-0.

**Correspondence and requests for materials** should be addressed to Kikuë Tachibana.

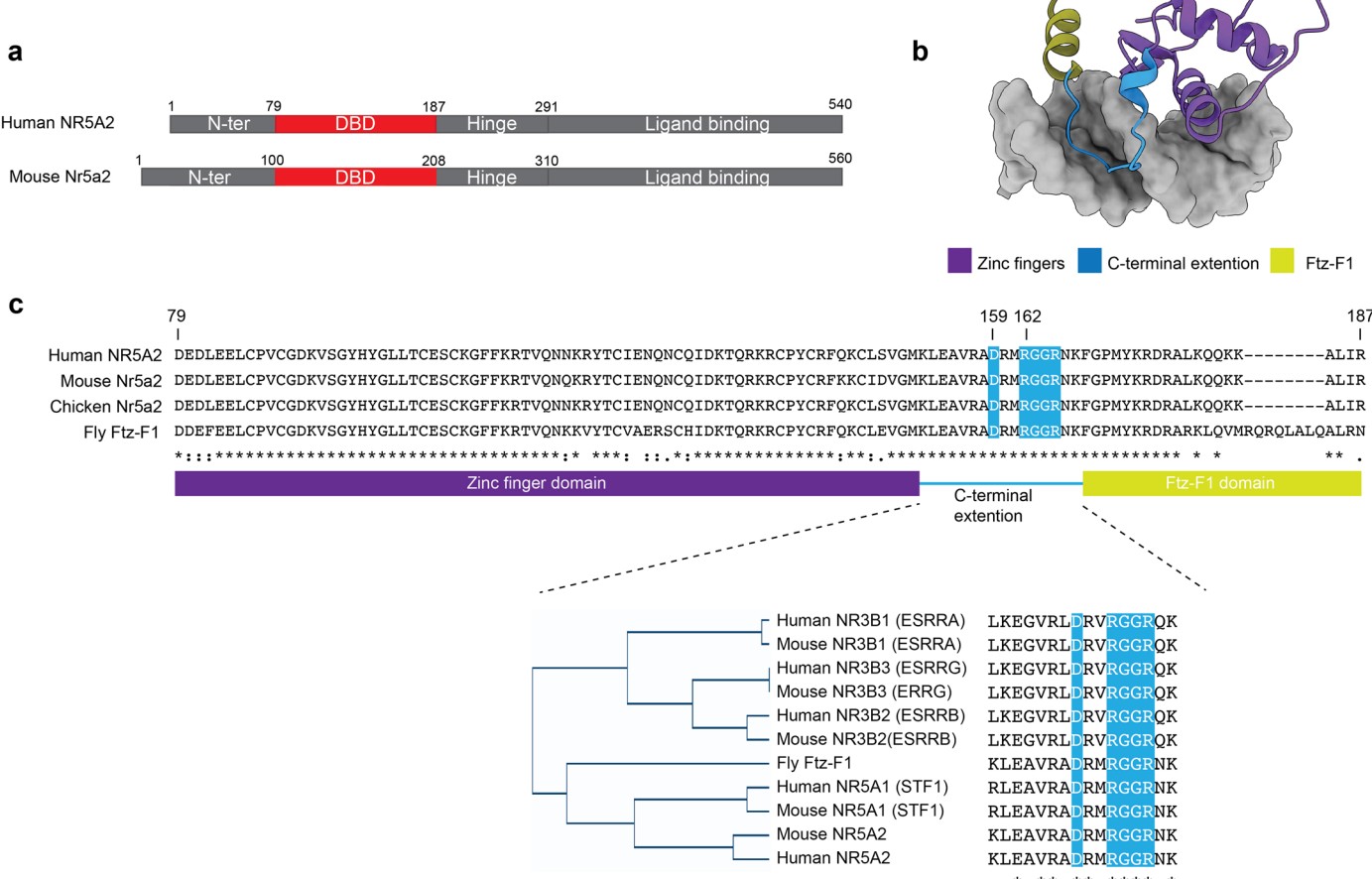

**Extended Data Fig. 1 | Sequence alignment of Nr5a2.** (**a**) Schematic representation showing the domains of mouse Nr5a2 and human NR5A2. Red squares indicate the DNA-binding domain. (**b**) The crystal structure of human NR5A2-DNA complex (PDBID: 2A66). Zinc-finger domains, C-terminal extension (CTE), and Ftz-F1 domain are shown in purple, blue, and yellow, respectively. (**c**) Sequence alignment showing DNA-binding domain of Nr5a2 (human, mouse, and chicken) and fly Ftz-F1 proteins. The number of amino acids indicates human NR5A2. The conservation of the CTE loop in NR5A and NR3B families is shown below.

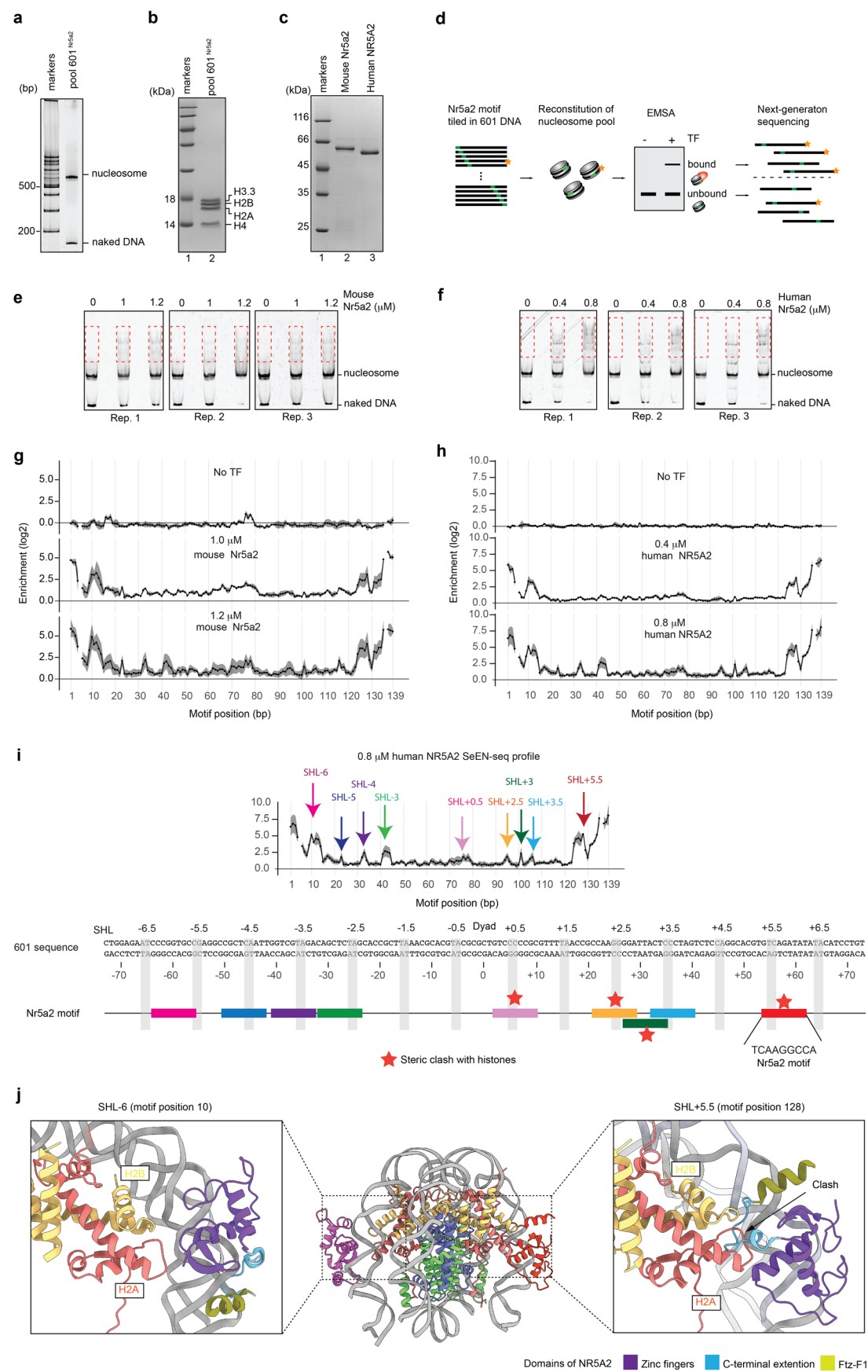

**Extended Data Fig. 2 | See next page for caption.**

**Extended Data Fig. 2 | SeEN-seq analysis. (a)** Purified nucleosome library containing Nr5a2 motif (pool 601^Nr5a2) was analyzed by native-PAGE with SYBR safe staining. Experiments were performed one time. **(b)** The histone contents of the purified nucleosome library were analyzed by 18% SDS-PAGE. Experiments were performed one time. **(c)** Purified mouse Nr5a2 (lane 2) and human NR5A2 (lane 3) used for SeEN-seq analysis. Lane1 indicates molecular markers. Experiments were performed one time. **(d)** Schematic illustration of SeEN-seq analysis. EMSA results were shown in panels E and F. **(e and f)** EMSA with mouse or human Nr5a2. Three independent experiments were performed. Red dot squares indicate sliced regions as bound fractions. **(g, h)** SeEN-seq enrichment profiles of mouse Nr5a2 and human NR5A2. The enrichments (log$_2$) were plotted against the positions where the NR5A2 motif (TCAAGGCCA) starts along the 601 DNA. The average values of three independent experiments are shown with the SD values. Motif positions #6 and #136 were lost due to technical issues. **(i)** Enriched sites identified by SeEN-seq analysis. Each color indicates the Nr5a2 motif in the different positions in the nucleosomal DNA. A representative SeEN-seq data with 0.8 μM human NR5A2 (n = 3, SD values) is shown. **(j)** The superimposed model of the canonical nucleosome bound by NR5A2 DBD. NR5A2 DBD does not show steric clash with histones at SHL -6 but not SHL + 5.5.

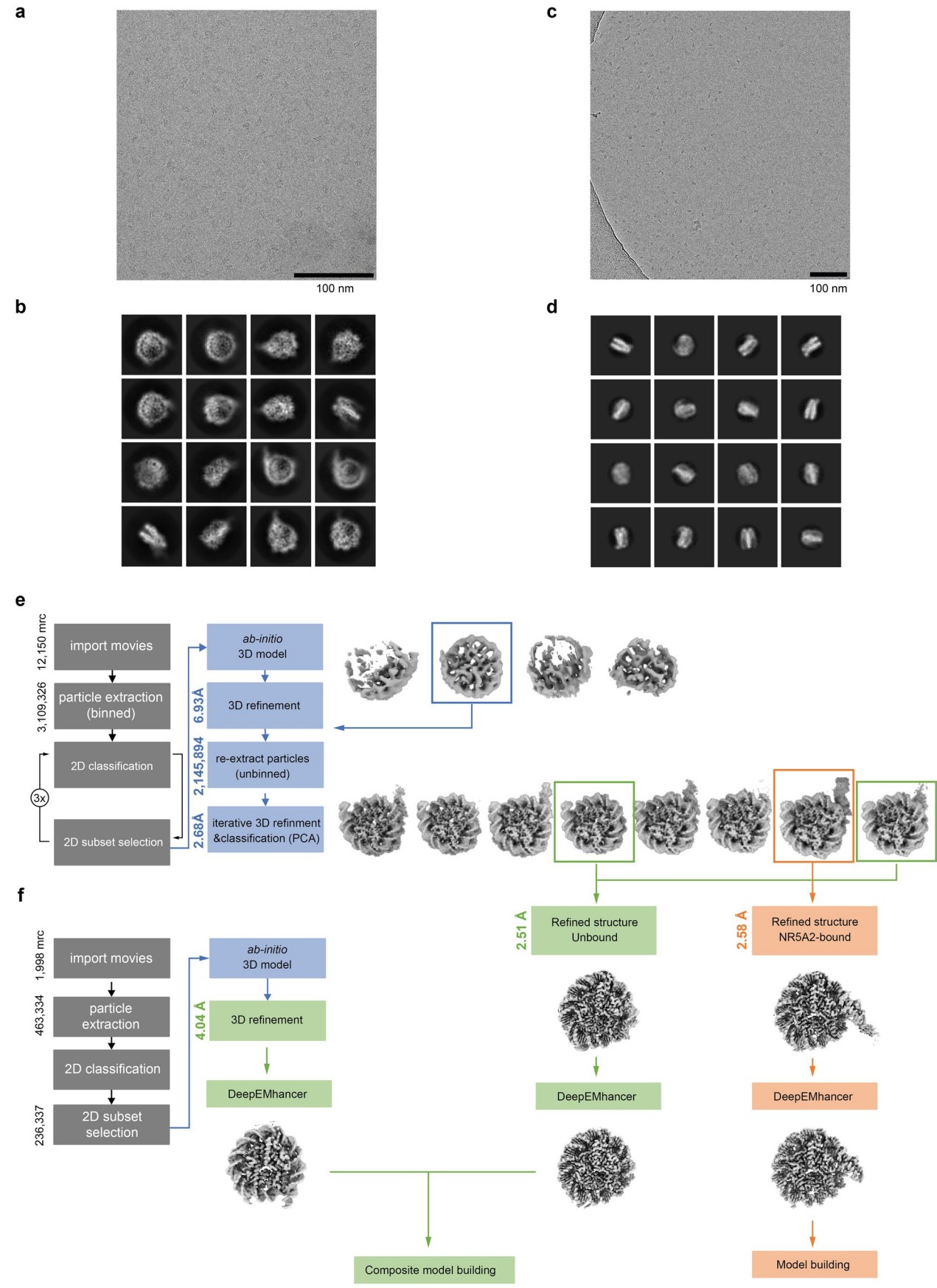

**Extended Data Fig. 3 | See next page for caption.**

**Extended Data Fig. 3 | Cryo-EM data processing flowchart for unbound and NR5A2-bound nucleosome.** (**a**) Digital micrograph of purified human NR5A2-nucleosome complex[SHL+5.5] acquired by Titan Krios G2. Similar micrographs were obtained at least three times. (**b**) Representative two-dimensional class averages from single particle images acquired by Titan Krios G2. (**c**) Digital micrograph of purified nucleosome containing Nr5a2 motif at SHL + 5.5 (601 nucleosome[SHL+5.5]) acquired by Glacios. Similar micrographs were obtained at least three times. (**d**) Representative two-dimensional class averages from single particle images acquired by Glacios. (**e**) Flowchart of the dataset obtained by Titan Krios G2. Cryo-EM maps of human NR5A2-nucleosome complex[SHL+5.5] and 601 nucleosome[SHL+5.5] were determined by CryoSPARC. (**f**) Flowchart of the dataset obtained by Glacios. 601 nucleosome[SHL+5.5] were determined by CryoSPARC live (version 4.1). Two EM maps were composited for the model building of unbound nucleosome.

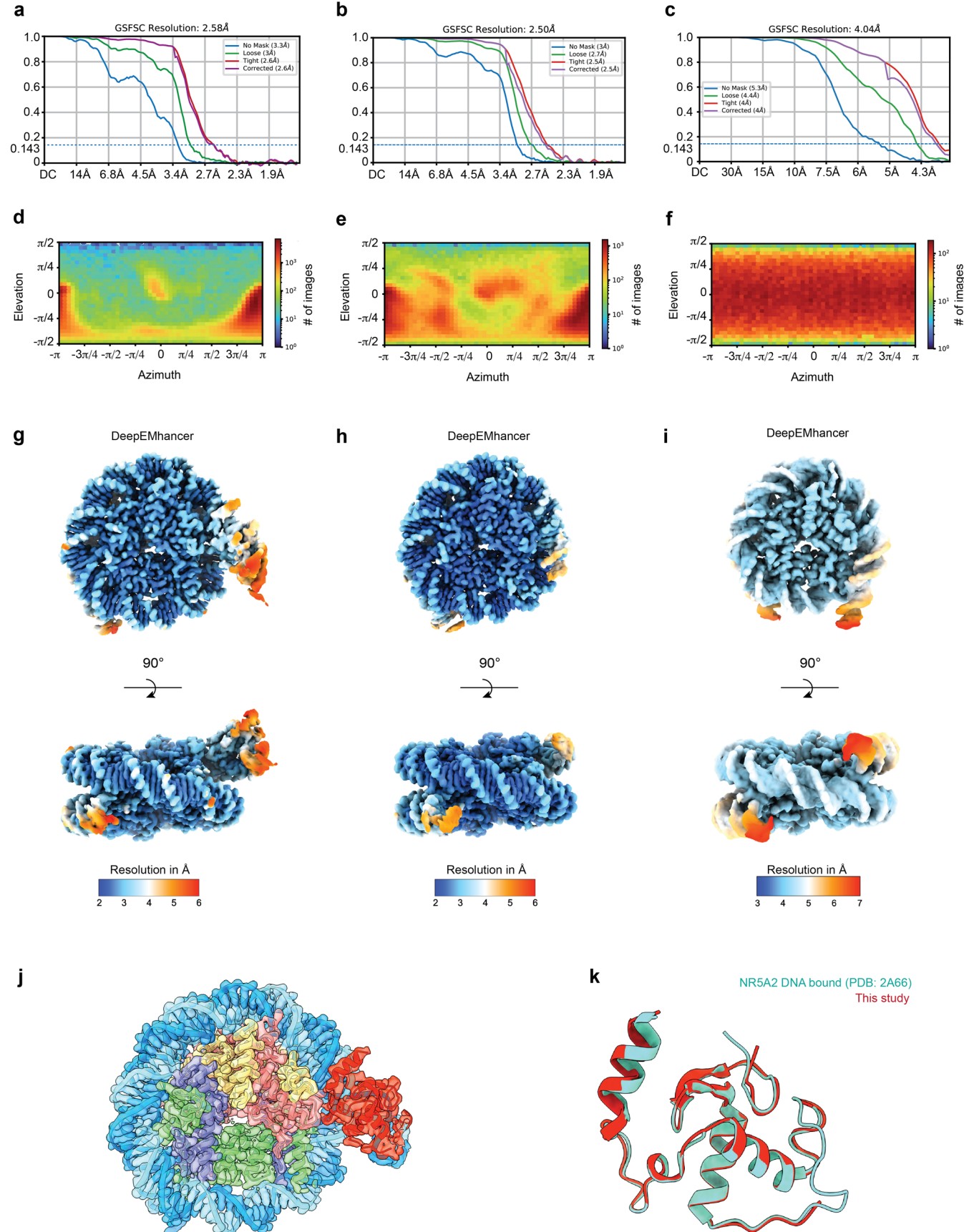

**Extended Data Fig. 4 | Data quality of NR5A2-nucleosome complex$^{SHL+5.5}$.**
(**a-c**) Fourier Shell Correlation (FSC) curves of maps with the FSC = 0.143 criterion.
(**d-f**) Angular distribution plot of particles employed to reconstruct maps.

(**g-i**) Local resolution of maps. (**j**) Model structure fitted to the map of NR5A2-nucleosome complex$^{SHL+5.5}$. (**k**) Superimposition of the human NR5A2 DBD bound to naked DNA (PDB: 2A66) and NR5A2 DBD bound to nucleosome (this study).

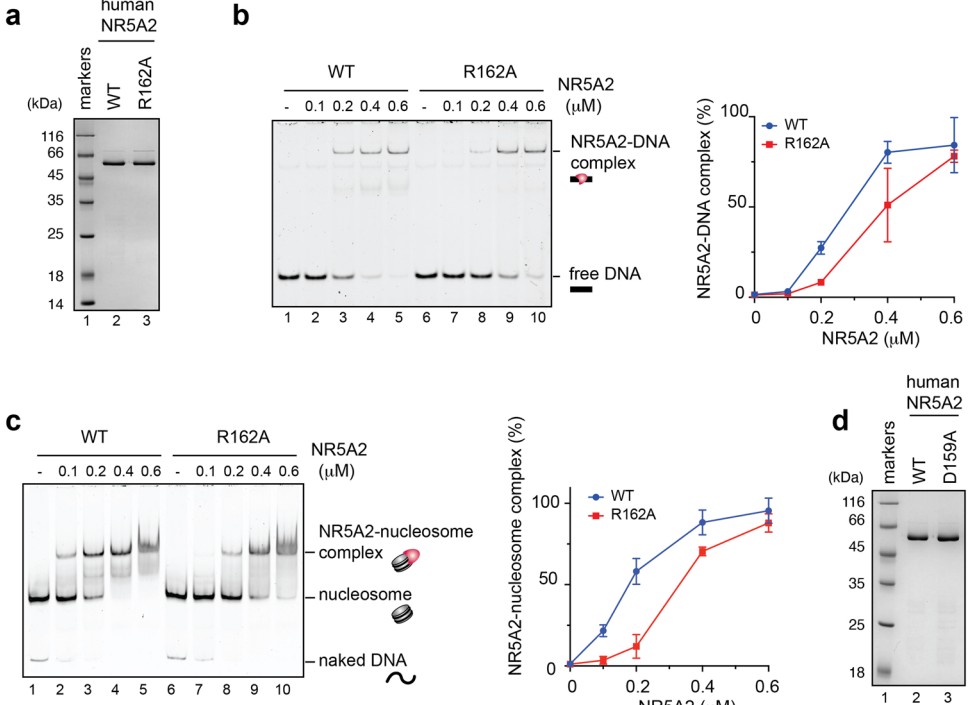

**Extended Data Fig. 5 | Purification of NR5A2 mutants and EMSA. (a)** Purified human NR5A2 wildtype (lane 2) and human NR5A2 R162A mutant (lane 3) were analyzed by 12% SDS-PAGE. Lane1 indicates molecular markers. Experiments were performed one time. (**b**) The left panels show representative data of EMSA with naked DNA containing Nr5a2 motif. Human NR5A2 wildtype (lanes 1-5) and NR5A2 R162A (lanes 6-10) were used for the experiments. Right panels show the graphical representations. The average values of three independent experiments are shown with the SD values. (**c**) The left panels show representative data of EMSA with 601 nucleosome containing Nr5a2 motif at SHL + 5.5. Human NR5A2 wildtype (lanes 1-5) and NR5A2 R162A (lanes 6-10) were used for the experiments. Right panels show the graphical representations. The average values of three independent experiments are shown with the SD values (**d**) Purified human NR5A2 D159A mutant (lane 3) were analyzed by 12% SDS-PAGE. Experiments were performed one time.

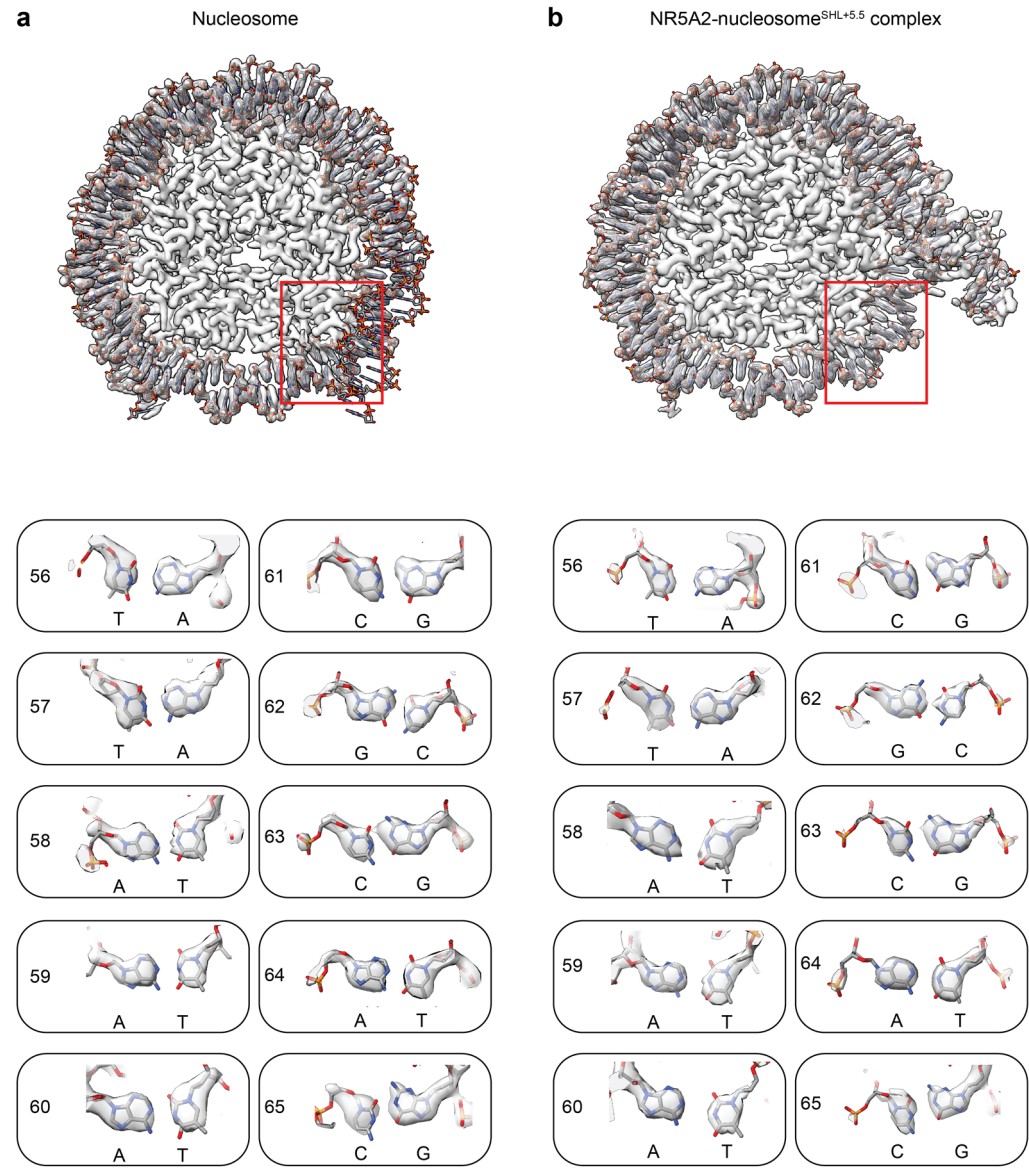

**a** Nucleosome

**b** NR5A2-nucleosome$^{SHL+5.5}$ complex

**Extended Data Fig. 6 | DNA registration assignment of the unbound and bound nucleosomes.** Illustrations showing DNA bases in the unbound (panel a) and bound (panel b) nucleosome structures. Cryo-EM density maps fit with each purine and pyrimidine at the same position of nucleosomal DNA.

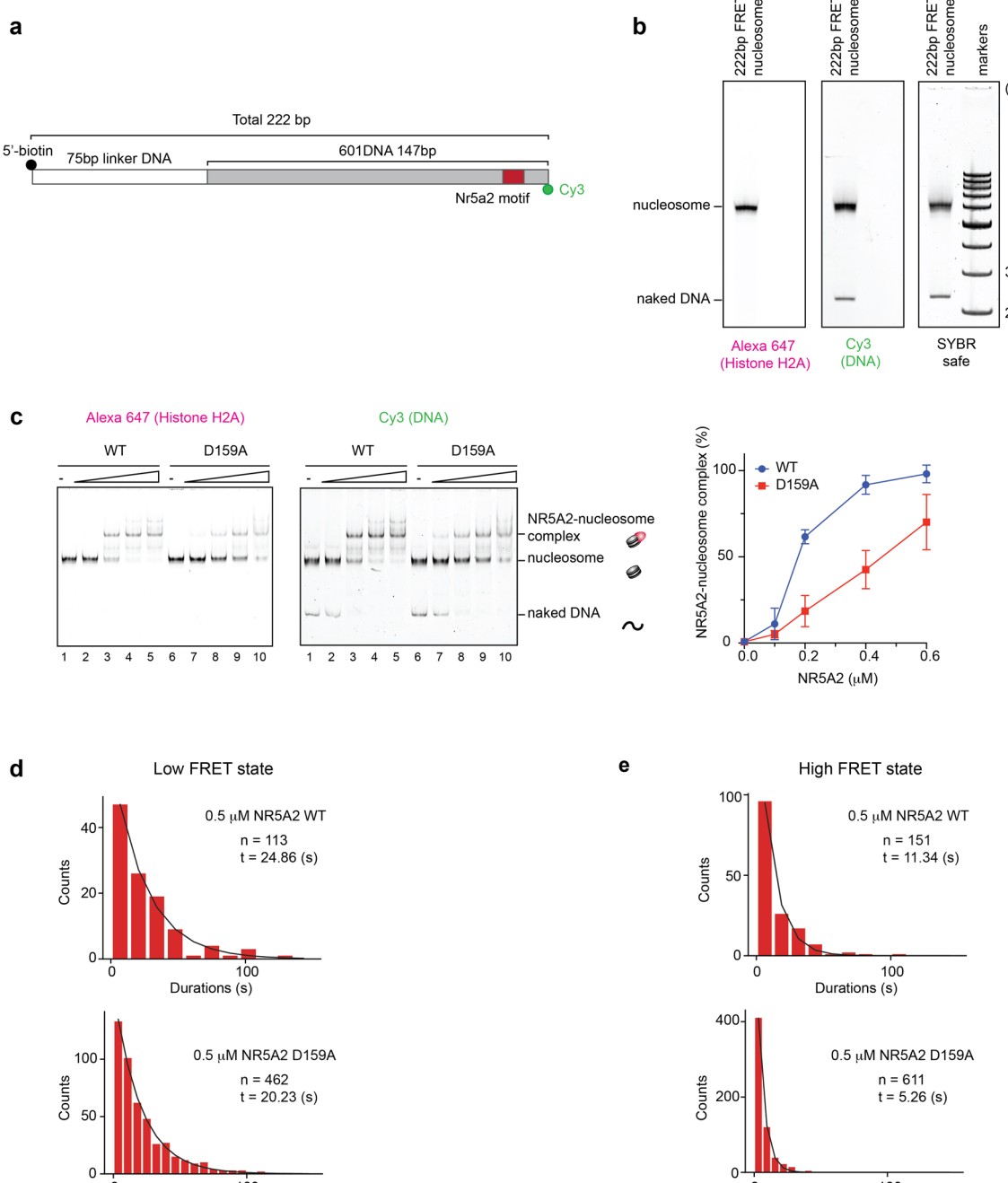

**Extended Data Fig. 7 | DNA design for single-molecule FRET analysis.**
(**a**) A 222 bp of DNA used for single-molecule FRET analysis. Biotin and Cy3 fluorescence dye were conjugated each 5′-DNA end. (**b**) Purified the nucleosomes for single-molecule FRET analysis. Nucleosomes were analyzed by 6% native-PAGE and detected by SYBR gold staining and Alexa 647 fluorescence. 100 bp markers were loaded in the right side of the sample. Experiments were performed one time. (**c**) The left panels show representative data of EMSA with the nucleosomes used for single-molecule FRET analysis. Alexa 647 fluorescence signals correspond to the nucleosome. The lanes 1-5 and lanes 6-10 show results with NR5A2 wildtype and NR5A2 D159A mutant, respectively. Right panels show the graphical representations. The average values of three independent experiments are shown with the SD values. (**d**) Histograms showing the duration in the low FRET state with the number of counts. (**e**) Histograms showing the duration in the high FRET state with the number of counts.

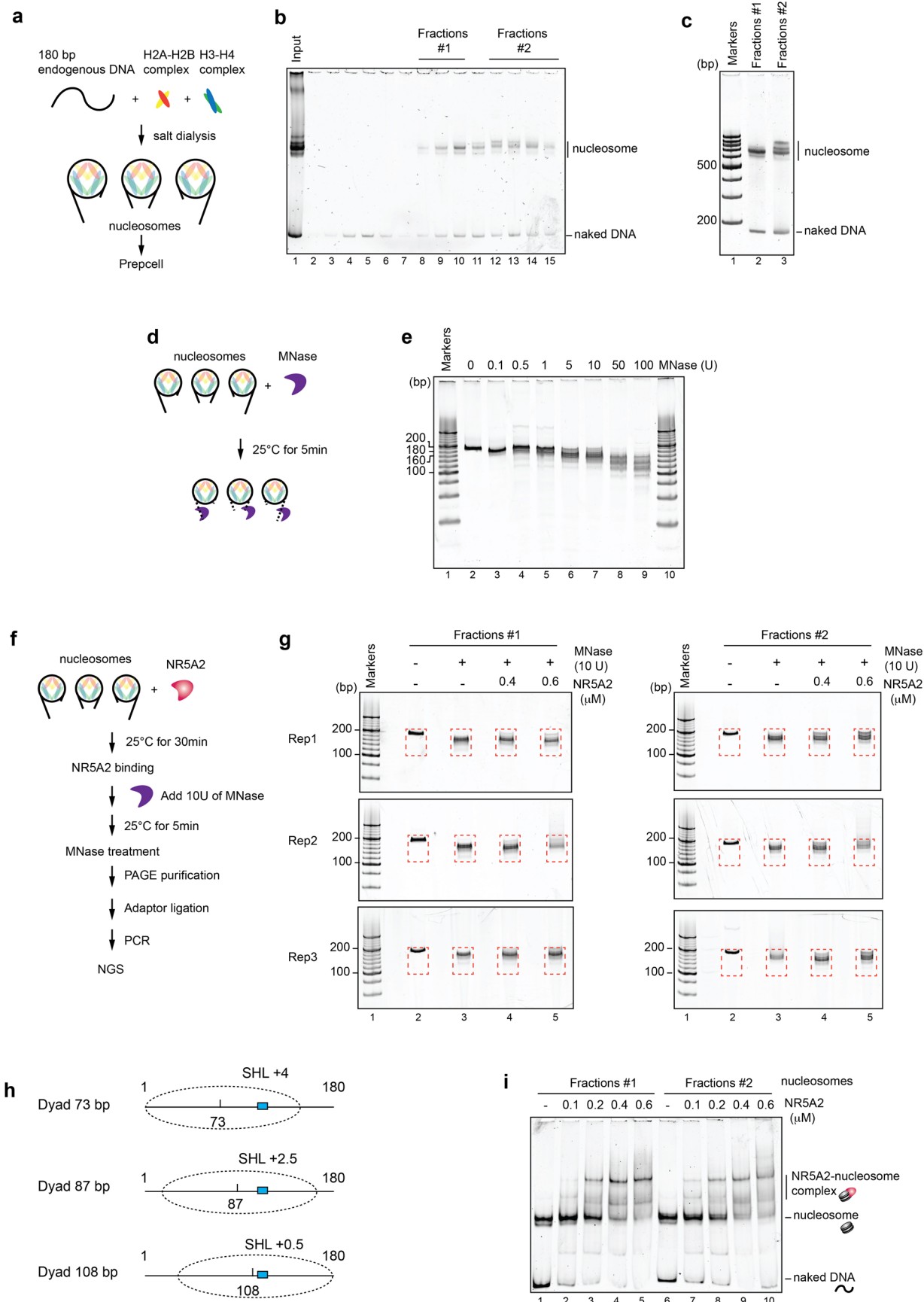

**Extended Data Fig. 8 | See next page for caption.**

**Extended Data Fig. 8 | MNase assay of 180 bp mouse endogenous nucleosome.** (**a**) An illustration showing nucleosome reconstitution with 180 bp DNA (**b**) Nucleosome purification by Prepcell apparatus. Lane numbers 8-10 and 12-15 were collected as fractions #1 and #2, respectively. Experiments were performed one time. (**c**) Purified nucleosomes were analyzed by native-PAGE with SYBR safe staining. Experiments were performed one time. (**d**) An illustration showing MNase treatment. Nucleosomal DNA is resistant to DNA cleavage by MNase. (**e**) MNase treatment of 180 bp mouse endogenous nucleosome. Lanes 1 and 10 indicate 20 bp DNA ladder. Experiments were performed one time. (**f**) An illustration showing MNase treatment in the presence of human NR5A2. MNase-resistant DNA fragments were ligated with adapter DNA and were sequenced. (**g**) MNase-resistant DNA fragments were native-PAGE with SYBR gold staining. Three independent experiments are shown. Red dot squares indicate sliced regions. (**h**) Nucleosome positions mapped by MNase assay. Blue squares indicate the Nr5a2 motif. (**i**) EMSA with 180 bp mouse endogenous nucleosome. The nucleosome containing fraction #1 (lanes 1-5) and fraction #2 (lanes 6-10) were used for the experiments. Human NR5A2 wild-type was titrated from 0 to 0.6 µM. Three independent experiments were performed, and the reproducibility was confirmed.

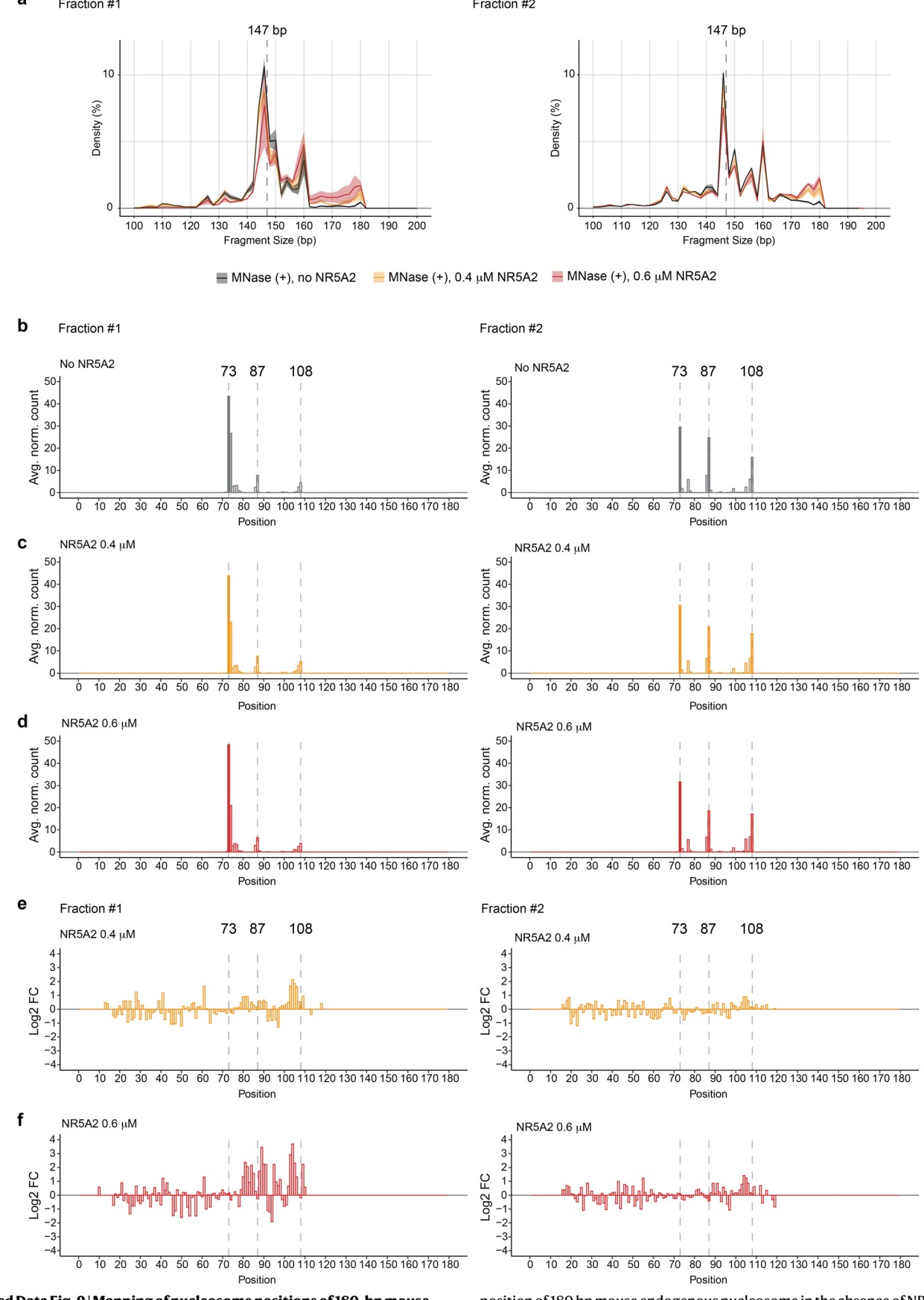

**Extended Data Fig. 9 | Mapping of nucleosome positions of 180 bp mouse endogenous genomic DNA. (a)** DNA fragment size distribution of nucleosomes containing each fraction (n = 3, SD values). **(b-d)** Histograms showing the dyad position of 180 bp mouse endogenous nucleosome in the absence of NR5A2 **(b)**, 0.4 μM NR5A2 **(c)**, and 0.6 μM NR5A2 **(d)**. **(e** and **f)** Log2-fold changes showing enrichments of the dyad positions of nucleosomes.

# Reporting Summary

## Statistics

For all statistical analyses, confirm that the following items are present in the figure legend, table legend, main text, or Methods section.

| n/a | Confirmed | |
|---|---|---|
| ☐ | ☒ | The exact sample size (*n*) for each experimental group/condition, given as a discrete number and unit of measurement |
| ☐ | ☒ | A statement on whether measurements were taken from distinct samples or whether the same sample was measured repeatedly |
| ☐ | ☒ | The statistical test(s) used AND whether they are one- or two-sided<br>*Only common tests should be described solely by name; describe more complex techniques in the Methods section.* |
| ☒ | ☐ | A description of all covariates tested |
| ☒ | ☐ | A description of any assumptions or corrections, such as tests of normality and adjustment for multiple comparisons |
| ☐ | ☒ | A full description of the statistical parameters including central tendency (e.g. means) or other basic estimates (e.g. regression coefficient) AND variation (e.g. standard deviation) or associated estimates of uncertainty (e.g. confidence intervals) |
| ☐ | ☒ | For null hypothesis testing, the test statistic (e.g. *F*, *t*, *r*) with confidence intervals, effect sizes, degrees of freedom and *P* value noted<br>*Give P values as exact values whenever suitable.* |
| ☒ | ☐ | For Bayesian analysis, information on the choice of priors and Markov chain Monte Carlo settings |
| ☒ | ☐ | For hierarchical and complex designs, identification of the appropriate level for tests and full reporting of outcomes |
| ☒ | ☐ | Estimates of effect sizes (e.g. Cohen's *d*, Pearson's *r*), indicating how they were calculated |

*Our web collection on statistics for biologists contains articles on many of the points above.*

## Software and code

Policy information about availability of computer code

| | |
|---|---|
| Data collection | EPU (Thermo Fisher Scientific) |
| Data analysis | Chimera X (Version 1.5), CryoSPARC, CryoSPARC Live (ver4.1), COOT, Phenix, Pymol, ImageLab (BIO-RAD), Prism9 (GraphPad), R, HOMER, Trim Galore, samtools (version 1.14), Picard (version 2.26.4), Bowtie2(version 2.3.5.1 and 2.4.4), deepTools, ggplot2, Mars (Molecule Archive Suite) |

For manuscripts utilizing custom algorithms or software that are central to the research but not yet described in published literature, software must be made available to editors and reviewers. We strongly encourage code deposition in a community repository (e.g. GitHub). See the Nature Portfolio guidelines for submitting code & software for further information.

## Data

Policy information about availability of data

All manuscripts must include a data availability statement. This statement should provide the following information, where applicable:
- Accession codes, unique identifiers, or web links for publicly available datasets
- A description of any restrictions on data availability
- For clinical datasets or third party data, please ensure that the statement adheres to our policy

The maps are available in the Electron Microscopy Data Bank (EMDB) database under accession no. EMD-17740 (NR5A2-nucleosome complex SHL+5.5) and EMD-17741 (601 nucleosome containing Nr5a2 motif at SHL+5.5). The atomic models are available in the Protein Data Bank database: PDB 8PKI (NR5A2-nucleosome complex SHL+5.5) and PDB 8PKJ (601 nucleosome containing Nr5a2 motif at SHL+5.5). PDBID: 3LZ0 and 2A66 were used for model building. PDBID:

1KX5 was used as canonical nucleosome structure. The dataset of Nr5a2 CUT&Tag and ATAC-seq were obtained from GSE17823423. The MNase data set for two-cell embryos were downloaded from the Genome Sequence Archieve (GSA) accession CRA00594433. The MNase data set for mESC was downloaded from the Gene Expression Omnibus (GEO) accession GSE8212740. Requests for plasmids generated in this study should be directed to the corresponding author.

## Research involving human participants, their data, or biological material

Policy information about studies with human participants or human data. See also policy information about sex, gender (identity/presentation), and sexual orientation and race, ethnicity and racism.

| | |
|---|---|
| Reporting on sex and gender | N/A |
| Reporting on race, ethnicity, or other socially relevant groupings | N/A |
| Population characteristics | N/A |
| Recruitment | N/A |
| Ethics oversight | N/A |

Note that full information on the approval of the study protocol must also be provided in the manuscript.

## Field-specific reporting

Please select the one below that is the best fit for your research. If you are not sure, read the appropriate sections before making your selection.

☒ Life sciences ☐ Behavioural & social sciences ☐ Ecological, evolutionary & environmental sciences

For a reference copy of the document with all sections, see nature.com/documents/nr-reporting-summary-flat.pdf

## Life sciences study design

All studies must disclose on these points even when the disclosure is negative.

| | |
|---|---|
| Sample size | No statistic method used to predetermine sample size. Sample sizes of 12,150 (Titan Krios) and 1,998 (Glacios) micrographs were sufficient for to provide enough particles for Cryo-EM map reconstruction. |
| Data exclusions | Junk particles were excluded in 2D and 3D classifications for Cryo-EM map reconstitution. |
| Replication | The reproducibility for all biochemical analyses was confirmed by at least three independent experiments. |
| Randomization | We have no data involving organisms and experiments that requires randomization. |
| Blinding | Blinding is not applicable for this study including structural and biochemical studies. |

## Reporting for specific materials, systems and methods

We require information from authors about some types of materials, experimental systems and methods used in many studies. Here, indicate whether each material, system or method listed is relevant to your study. If you are not sure if a list item applies to your research, read the appropriate section before selecting a response.

### Materials & experimental systems

| n/a | Involved in the study |
|---|---|
| ☒ | Antibodies |
| ☒ | Eukaryotic cell lines |
| ☒ | Palaeontology and archaeology |
| ☒ | Animals and other organisms |
| ☒ | Clinical data |
| ☒ | Dual use research of concern |
| ☒ | Plants |

### Methods

| n/a | Involved in the study |
|---|---|
| ☒ | ChIP-seq |
| ☒ | Flow cytometry |
| ☒ | MRI-based neuroimaging |

