## [Peer Review File · Nature Structural & Molecular Biology]

Peer Review Information

Manuscript Title: Nucleosome-bound NR5A2 structure reveals pioneer factor mechanism by minor groove anchor competition

Corresponding author name(s): Kikuë Tachibana

Reviewer Comments & Decisions:

Decision Letter, initial version:
--

Message: 15th Aug 2023

Dear Dr. Tachibana,

Thank you again for submitting your manuscript "Nucleosome-bound NR5A2 structure reveals pioneer factor mechanism by minor groove anchor competition". I apologize for the delay in responding, which resulted from the difficulty in obtaining suitable referee reports. Nevertheless, we now have comments (below) from the 2 reviewers who evaluated your paper. In light of those reports, we remain interested in your study and would like to see your response to the comments of the referees, in the form of a revised manuscript.

You will see that all reviewers appreciate the results and find the conclusions timely and of wide interest. There are, however, several concerns and suggestions that should be addressed in a revision. Specifically, Reviewer #2 suggests that you further explore NR5A2 binding in open vs. closed chromatin, which we agree would strengthen the study. In addition, both Reviewers raise a number of important points that require clarification. Additionally, please be sure to include the CryoEM maps and models in the revised submissions, for the referees to assess.

Please be sure to address/respond to all concerns of the referees in full in a point-by-point response and highlight all changes in the revised manuscript text file. If you have comments that are intended for editors only, please include those in a separate cover letter.

We expect to see your revised manuscript within 3-4 months. If you cannot send it within this time, please contact us to discuss an extension; we would still consider your revision, provided that no similar work has been accepted for publication at NSMB or published elsewhere.

Reporting Summary:

Please note that all key data shown in the main figures as cropped gels or blots should be presented in uncropped form, with molecular weight markers. These data can be aggregated into a single supplementary figure item. While these data can be displayed in a relatively informal style, they must refer back to the relevant figures. These data should be submitted with the final revision, as source data, prior to acceptance, but you may want to start putting it together at this point.

SOURCE DATA: we urge authors to provide, in tabular form, the data underlying the graphical representations used in figures. This is to further increase transparency in data reporting, as detailed in this editorial (<http://www.nature.com/nsmb/journal/v22/n10/full/nsmb.3110.html>). Spreadsheets can

be submitted in excel format. Only one (1) file per figure is permitted; thus, for multi-paneled figures, the source data for each panel should be clearly labeled in the Excel file; alternately the data can be provided as multiple, clearly labeled sheets in an Excel file. When submitting files, the title field should indicate which figure the source data pertains to. We encourage our authors to provide source data at the revision stage, so that they are part of the peer-review process.

Data availability: this journal strongly supports public availability of data. All data used in accepted papers should be available via a public data repository, or alternatively, as Supplementary Information. If data can only be shared on request, please explain why in your Data Availability Statement, and also in the correspondence with your editor. Please note that for some data types, deposition in a public repository is mandatory - more information on our data deposition policies and available repositories can be found below: <https://www.nature.com/nature-research/editorial-policies/reporting-standards#availability-of-data>

Nature Structural & Molecular Biology is committed to improving transparency in authorship. As part of our efforts in this direction, we are now requesting that all authors identified as 'corresponding author' on published papers create and link their Open Researcher and Contributor Identifier (ORCID) with their account on the Manuscript Tracking System (MTS), prior to acceptance. This applies to primary research papers only. ORCID helps the scientific community achieve unambiguous attribution of all scholarly contributions. You can create and link your ORCID from the home page of the MTS by clicking on 'Modify my Springer Nature account'. For more information please visit please

visit <http://www.springernature.com/orcid>

[Redacted]

Sincerely,

Carolina

Carolina Perdigoto, PhD
Chief Editor
Nature Structural & Molecular Biology
orcid.org/0000-0002-5783-7106

Reviewers' Comments:

Reviewer #1:

Remarks to the Author:

The field of pioneer factors has moved from straightforward assays of assessing how well transcription factors bind nucleosomes to detailed structure and crosslinking studies that reveal pioneer factors interacting with histones on the nucleosome and modulating nucleosome structure. Perhaps the most interesting aspect of the recent studies is that there are several different ways that pioneer factors can interact with histones and modulate nucleosome structure, and this paper provides evidence for a minor groove displacement model. All of these details are important because eventually the information will be used to make "designer" pioneer factors where we more specifically open up silent genes to change cell states and fates. The authors and others have provided substantial genetic, genomic, and biochemical evidence that the nuclear receptor family member NR5A2 binds nucleosomes and acts as a pioneer factor. NR5A2 binds as a monomer, which is unusual for nuclear receptors but may have enabled the facility with which the authors were able to study NR5A2 bound to a nucleosome via cryoEM. They tiled an NR5A2 site across a Widom 601 nucleosome sequence. Like all other studies employing that sequence, they find that their factor binds the entry and exit sites of a nucleosome. Since the few (unfortunately) studies that use natural or SELEX-derived sequences are finding binding at various sites across the nucleosome, the present study is somewhat limited by using the 601. However, to their credit, the also authors studied a natural sequence nucleosome which did not enable structure analysis but could gain evidence that NR5A2 could bind at SH+2 site on such nucleosomes (as opposed to the SH 5.5 site seen here and all other 601 studies). Regardless of this point, the authors discerned that NR5A2 displaces a DNA minor groove interaction that the nucleosome normally has with histone H2A Arg 77, substituting with an interaction with NR5A2 R162. This only moderately but detectably affects DNA and nucleosome binding by the factor with tweaking of the histone backbone geometries that face the side where NR5A2 bind. More significantly, they found Asp159 of NR5A2 seems to clash with an H2A loop K75, forcing K75 to flip and H bond with H2A His82, with a shift in the $\alpha 2$ helix of H2A. NR5A2 binding also led to a shift in H3; thus NR5A2 binding, which helps release DNA at the end of the nucleosome, is doing so via various interactions with the histones and changes in local histone structure. In sum, these observations contribute substantially to our understanding of the impact of pioneer factor binding to nucleosomes. I have a few comments as noted below, but in general the study is well controlled and makes an excellent contribution to NSMB.

1. The results section ends on an unexplained result for which I may have an explanation.

The NR5A2 D159A mutant has an interesting phenotype on target nucleosomes in a FRET assay, which is that it exhibits a short duration of high FRET (unbound) states (Fig. 5C/D), implying that the factor has a higher on-rate. Yet the factor clearly has a shorter duration of low FRET (bound) states, concordant with the EMSA data (Fig. 4D). By having fewer bound states, it generates more free protein which could allow for the appearance of higher on-rates.

2. Various of the figures are not colored or illustrated well enough to be easily interpreted. Fig. 2D, should use 2-colors rather than 3 to highlight differences in RMSDs. Fig. 3, not easy to see the lysine amino acid flip in panel b with the blobs as shown. Fig. 4f, please make the NR5A2 protein strand a color that does not blend so much with the purple DNA.

3. The NR5A2 D159A protein looks like it has some deficiency in free DNA binding (Fig. 3) though clearly more so on nucleosomes. It is better for the authors to acknowledge a minor deficit of free DNA binding.

4. page 8, line 3, should say "wild type and D159A showed similar relative binding affinity" (add word "relative")

Reviewer #2:

Remarks to the Author:

The manuscript nicely presents the structure of the transcription factor NR5A2, bound to its nucleosome-embedded motif. The authors identify that a conserved C-terminal extension (CTE) region between the zinc finger and Ftz-F1 domain is important for nucleosome binding at a high-affinity position (SHL+5.5), as determined by SeEN-seq. The binding of NR5A2 at this DNA position competes with histone-DNA interactions within the minor groove, and mutation of a conserved residue, D159A, reduces nucleosome affinity, both for the W601-SHL+5.5 substrate and a native DNA sequence, where the motif is likely more internal.

The structure and accompanying biochemistry are very thorough and support the conclusions. The results will be of interest to many in the field of chromatin and nuclear receptor biology. I only have a few major and minor points listed below:

Major points:

- The finding that the C-terminal extension region that interacts with the minor groove of DNA is important to compete with histones for DNA, enabling nucleosome binding, is very interesting. This suggests that the affinity of the CTE for the underlying DNA sequence or motif could also influence its nucleosome binding. I wonder if the DNA sequence in the "canonical" high-affinity motif in the TCA extension region (TCAAGGCCA) is required for nucleosome or 'closed chromatin' binding in vivo, while other, accessible regions of the genome i.e. 'open chromatin' the motif could be more flexible. Now with your structure in hand, could you go back to the genomic data you describe in your 2022 Science paper: DOI: 10.1126/science.abn747 and examine NR5A2 binding in open vs. closed chromatin (by ATAC-seq?) and compare the motif enrichment profiles? I would expect that you might find that the motif logo is more permissive in open chromatin, particularly within the region that contacts the CTE. In contrast, the high-affinity motif may be strictly required in nucleosome-occupied "closed" regions. If this is not possible, another structure-informed cellular data analysis would strengthen the manuscript.

- In Fig. 2, a comparison is made between the unbound and NR5A-bound nucleosome models, and an RMSD is calculated between the C-alpha of the histone octamers (also Page 6). This calculation is extended to the 'bound' side of histones, near the NR5A2 binding site, and the 'unbound' side, and shows a minor change near the 'bound' side, and the authors suggest this could be due to NR5A2 binding / minor groove competition. How was the register or DNA sequence in the unbound nucleosome determined to align the structures? Was the map of sufficient resolution to identify the DNA bases? If the DNA is not unambiguously assigned, which breaks the particle's symmetry, the histone octamer could have been aligned inversely, possibly giving rise to this slight difference in the C-alpha conformations. If the DNA register/symmetry in the structure was determined, the conformational shifts are quite minimal and could arise from different model-building procedures/sample conditions, so I would carefully interpret these changes as a direct result of NR5A2 binding.

- When examining NR5A2 binding to the native DNA sequence in vivo, how precise does the Mnase experiment in cells report on nucleosome position in this bulk assay? From the scale bar (Fig.4a) and it's a bit hard to see, and it seems there may be a bit of a distribution of peaks or nucleosome positions across the DNA fragment, as is common in mammalian cells. It would be nice in the current manuscript to detail the cellular experimental design and interpretation a bit more so that the reader can understand where the sequence was derived. On a related note, to identify the exact register of the native DNA sequence in vitro, would it be possible to take a longer fragment from this locus, ~180bp (containing the 149bp you selected as the peak nucleosome-fragment), and perform Mnase + sequencing in vitro +/- NR5A2? I think this experiment, if possible, could help identify more precisely the DNA register of the nucleosome in vitro, which may place the motif at SHL2.5, as predicted if the 149bp fragment had the energetically most favorable dyad position in the middle of the DNA sequence. Still, it could also have many different registers/translational positions on this non-601 native fragment, as described in other native sequences for CENP-B, quite some time ago:

<https://doi.org/10.1074/jbc.M509666200> or more recently for OCT4:

<https://doi.org/10.1038/s41586-023-06112-6>, just to list a few examples. The addition of NR5A2 and comparison of the "bound" and "unbound" registers could show if NR5A2 positions a nucleosome, like seen with other DNA-binding proteins, or, since it doesn't appear to make direct histone contacts, but instead competes with histones for DNA, it may not influence the rotational/translational nucleosome position.

- While the interactions reported here with a nuclear receptor and DNA in the presence of histones are novel, I wanted to point out that minor groove competition with the histones has also been observed with SOX proteins: <https://doi.org/10.1038/s41586-020-2195-y> where "SOX11 widens the DNA minor groove.. and pulls the DNA away from the histone octamer".

Minor points

- In Figure 1, panel a, I would suggest to include in the SHL motif positions in the X-axis of the SeEN-seq plot, as this is the nomenclature used throughout the manuscript.
- In the cryo-EM processing figures, it would be good to show a raw micrograph from each dataset to judge the particle distribution and a selection of 2Ds, as 2D classification was mentioned in the processing pipeline.
- In the EXD Fig. 3 legend, the datasets are referred to as the "Krios" and "Glacios" datasets. I would also suggest including the names of the corresponding samples here. It would also be helpful to list in the legend which software was used, in addition to the Methods.
- In the manuscript, it is stated that the "NR52-nucleosome complex showed that the side

chain of H2A Lys75 (loop 2) was flipped and formed a new hydrogen bond with HA His82 (Fig 3a).” Is the density in the cryoEM map resolved well enough to see this side chain flip? With such a detailed analysis, it would be good to show the corresponding density in a zoomed view in both the bound and unbound complexes in the Figure.

- When referencing other TF-nucleosome structure papers, including OCT4 and SOX (Page 3), the BioRxiv paper has been published: <https://doi.org/10.1038/s41586-023-06112-6> and I would also include this recent paper as well: <https://doi.org/10.1038/s41586-023-06112-6>

- In 4e, I thought it would be helpful to indicate with colors or pointers the specific regions of NR5A2, such as ZF, CTE, etc. so that one can see the CTE in this instance would also be modeled to face the histones, like in the SHL+5.5 instance.

Author Rebuttal to Initial comments

We would like to sincerely thank all reviewers for their positive comments and constructive suggestions, which we have fully addressed in the revised manuscript. For clarity, we have highlighted new data and text in yellow in the manuscript. Our work provides structural insights into DNA minor groove competition by pioneer factor NR5A2 to destabilize nucleosomes and facilitate gene expression. The NR5A2-nucleosome structure provides the first evidence of how an orphan nuclear receptor directly competes with core histone-DNA interactions to cause unwrapping of DNA, which is a crucial step to generate accessible chromatin. Our findings are important for understanding how Nr5a2 controls genome-wide activation of the zygotic genome (Gassler et al., 2022), greatly potentiates induced reprogramming to pluripotency and contributes to cancer.

The revision experiments suggested by reviewers have greatly strengthened the manuscript. In brief:

- 1) The interpretation offered by reviewer 1 on the smFRET results stimulated us to perform additional experiments to better understand the on- and off-rates of NR5A2 D159A mutant binding to nucleosomes. Our new results based on a larger sample size showed that the D159A mutation on the CTE loop significantly affects the off-rate of nucleosome binding (**new Fig, 4e-g**).
- 2) As suggested by reviewer 2, we examined Nr5a2 CUT&Tag and ATAC-seq data to determine whether the “TCA” motif extension suggested by the structural work has implications for NR5A2 binding to closed chromatin. We found that the “TCA” extension is highly enriched in closed chromatin and slightly permissive in open chromatin. It is therefore plausible that Nr5a2 requires minor groove competition for nucleosome binding on nucleosome-occupied “closed chromatin” (**new Fig. 2d and 2e**).
- 3) As suggested by reviewer 2, we performed MNase-seq to map nucleosome positioning on the endogenous *SINE B1* sequence. NR5A2 binds to differently positioned nucleosomes that contain the Nr5a2 motif at SHL +0.5, +2.5 or +4 without inducing a shift in register (**new Extended Data Figs. 8 and 9**). We conclude that NR5A2 can bind to its motif in the context of the nucleosome at several positions.
- 4) The DNA bases were unambiguously assigned to the unbound nucleosome from the cryo-EM map density (**new Extended Data Fig. 6**).

Please find our detailed responses below.

Reviewer #1

Remarks to the Author:

The field of pioneer factors has moved from straightforward assays of assessing how well transcription factors bind nucleosomes to detailed structure and crosslinking studies that reveal pioneer factors interacting with histones on the nucleosome and modulating nucleosome structure. Perhaps the most interesting aspect of the recent studies is that there are several different ways that pioneer factors can interact with histones and modulate nucleosome structure, and this paper provides evidence for a minor groove displacement model. All of these details are important because eventually the information will be used to make "designer" pioneer factors where we more specifically open up silent genes to change cell states and fates. The authors and others have provided substantial genetic, genomic, and biochemical evidence that the nuclear receptor family member NR5A2 binds nucleosomes and acts as a pioneer factor. NR5A2 binds as a monomer, which is unusual for nuclear receptors but may have enabled the facility with which the authors were able to study NR5A2 bound to a nucleosome via cryoEM. They tiled an NR5A2 site across a Widom 601 nucleosome sequence. Like all other studies employing that sequence, they find that their factor binds the entry and exit

sites of a nucleosome. Since the few (unfortunately) studies that use natural or SELEX-derived sequences are finding binding at various sites across the nucleosome, the present study is somewhat limited by using the 601. However, to their credit, the also authors studied a natural sequence nucleosome which did not enable structure analysis but could gain evidence that NR5A2 could bind at SH+2 site on such nucleosomes (as opposed to the SH 5.5 site seen here and all other 601 studies). Regardless of this point, the authors discerned that NR5A2 displaces a DNA minor groove interaction that the nucleosome normally has with histone H2A Arg 77, substituting with an interaction with NR5A2 R162. This only moderately but detectably affects DNA and nucleosome binding by the factor with tweaking of the histone backbone geometries that face the side where NR5A2 bind. More significantly, they found Asp159 of NR5A2 seems to clash with an H2A loop K75, forcing K75 to flip and H bond with H2A His82, with a shift in the $\alpha 2$ helix of H2A. NR5A2 binding also led to a shift in H3; thus NR5A2 binding, which helps release DNA at the end of the nucleosome, is doing so via various interactions with the histones and changes in local histone structure. In sum, these observations contribute substantially to our understanding of the impact of pioneer factor binding to nucleosomes. I have a few comments as noted below, but in general the study is well controlled and makes an excellent contribution to NSMB.

We are very grateful to the reviewer for highlighting our findings and would like to sincerely thank the reviewer for supporting the work.

1. The results section ends on an unexplained result for which I may have an explanation. The NR5A2 D159A mutant has an interesting phenotype on target nucleosomes in a FRET assay, which is that it exhibits a short duration of high FRET (unbound) states (Fig. 5C/D), implying that the factor has a higher on-rate. Yet the factor clearly has a shorter duration of low FRET (bound) states, concordant with the EMSA data (Fig. 4D). By having fewer bound states, it generates more free protein which could allow for the appearance of higher on-rates.

We thank the reviewer for pointing out this possible explanation that we had not considered previously. In our smFRET assay, we applied 100 nM NR5A2 to flow cells containing approximately 400 pM nucleosomes bound to the surface of the cover slip for imaging. We therefore have a 250x excess of NR5A2 over nucleosomes. If all protein would unbind at the same moment, the overall protein content would increase by 0.4 %. Based on this estimate, the contribution of additional free protein to the on-rate during our observations is expected to be negligible.

Upon further examination of our results, we noticed that the number of molecules observed for NR5A2 D159A was not comparable to wildtype. Therefore, we performed three independent experiments to further increase our observation numbers. Consistent with the results reported in the initial submission, our revised value for the duration of the low FRET state for NR5A2 D159A is on average shorter compared to wild-type NR5A2 (NR5A2 wild-type: 16.4 ± 1.57 s, NR5A2 D159A: 9.19 ± 0.54 s) (**new Fig, 4e and 4g**). In contrast, our revised values for the average lifetimes of the high FRET states are of similar duration (NR5A2 wild-type: 20.27 ± 2.51 s, NR5A2 D159A: 16.44 ± 2.92 s) (**new Fig, 4f and 4g**). Consistent with the results reported in the initial submission, the high FRET states for NR5A2 D159A are on average slightly shorter in duration, however, the difference is within the uncertainty of our measurements. We therefore conclude the mutation on the CTE loop significantly affects only the off-rate of nucleosome binding.

2. Various of the figures are not colored or illustrated well enough to be easily interpreted. Fig. 2D, should use 2-colors rather than 3 to highlight differences in RMSDs. Fig. 3, not easy to see the lysine amino acid flip in panel b with the blobs as shown. Fig. 4f, please make the NR5A2 protein strand a color that does not blend so much with the purple DNA.

Thank you for the suggestions. We changed to a 2-color scheme with a narrow range of RMSD (**new Fig. 2f**). We now show flipped H2A Lys 75 with a cryo-EM density map (**new Fig. 3a**). We also colored the Nr5a2 motif DNA in cyan for better visualization (**new Fig, 5b and 5c**).

3. The NR5A2 D159A protein looks like it has some deficiency in free DNA binding (Fig. 3) though clearly more so on nucleosomes. It is better for the authors to acknowledge a minor deficit of free DNA binding.

We agree and have amended this (page 7, lane 10).

4. page 8, line 3, should say "wild type and D159A showed similar relative binding affinity" (add word "relative").

We added the word for clarity in the manuscript (page 7, lane 25).

Reviewer #2:

Remarks to the Author:

The manuscript nicely presents the structure of the transcription factor NR5A2, bound to its nucleosome-embedded motif. The authors identify that a conserved C-terminal extension (CTE) region between the zinc finger and Ftz-F1 domain is important for nucleosome binding at a high-affinity position (SHL+5.5), as determined by SeEN-seq. The binding of NR5A2 at this DNA position competes with histone-DNA interactions within the minor groove, and mutation of a conserved residue, D159A, reduces nucleosome affinity, both for the W601-SHL+5.5 substrate and a native DNA sequence, where the motif is likely more internal.

The structure and accompanying biochemistry are very thorough and support the conclusions. The results will be of interest to many in the field of chromatin and nuclear receptor biology. I only have a few major and minor points listed below:

We sincerely thank the reviewer for providing helpful suggestions that have resulted in a stronger manuscript.

Major points:

- The finding that the C-terminal extension region that interacts with the minor groove of DNA is important to compete with histones for DNA, enabling nucleosome binding, is very interesting. This suggests that the affinity of the CTE for the underlying DNA sequence or motif could also influence its nucleosome binding. I wonder if the DNA sequence in the "canonical" high-affinity motif in the TCA extension region (TCAAGGCCA) is required for nucleosome or 'closed chromatin' binding *in vivo*, while other, accessible regions of the genome i.e. 'open chromatin' the motif could be more flexible. Now with your structure in hand, could you go back to the genomic data you describe in your 2022 Science paper: DOI: 10.1126/science.abn747 and examine NR5A2 binding in open vs. closed chromatin (by ATAC-seq?) and compare the motif enrichment profiles? I would expect that you might find that the motif logo is more permissive in open chromatin, particularly within the region that contacts the CTE. In contrast, the high-affinity motif may be strictly required in nucleosome-occupied "closed" regions. If this is not possible, another structure-informed cellular data analysis would strengthen the manuscript.

We thank the reviewer for suggesting this important additional analysis. To address how the affinity of the DNA motif recognized by the CTE loop influences nucleosome binding *in vivo*, we performed *de novo* motif

searches using Nr5a2 CUT&Tag and ATAC-seq data in mouse 2-cell embryos (Gassler et al., 2022). As predicted by the reviewer, we found that the “TCA” extension in the motif is slightly permissive in the open chromatin that includes nucleosome-depleted regions (**new Fig. 2e**). In contrast, the “TCA” extension, especially the third adenine, is highly enriched in closed chromatin bound by Nr5a2 (**new Fig. 2e**). It is therefore plausible that Nr5a2 requires minor groove competition for nucleosome binding on nucleosome-occupied “closed chromatin”. This analysis further supports a model for minor groove competition via the CTE loop for nucleosome binding.

• In Fig. 2, a comparison is made between the unbound and NR5A-bound nucleosome models, and an RMSD is calculated between the C-alpha of the histone octamers (also Page 6). This calculation is extended to the ‘bound’ side of histones, near the NR5A2 binding site, and the ‘unbound’ side, and shows a minor change near the ‘bound’ side, and the authors suggest this could be due to NR5A2 binding / minor groove competition. How was the register or DNA sequence in the unbound nucleosome determined to align the structures? Was the map of sufficient resolution to identify the DNA bases? If the DNA is not unambiguously assigned, which breaks the particle's symmetry, the histone octamer could have been aligned inversely, possibly giving rise to this slight difference in the C-alpha conformations. If the DNA register/symmetry in the structure was determined, the conformational shifts are quite minimal and could arise from different model-building procedures/sample conditions, so I would carefully interpret these changes as a direct result of NR5A2 binding.

We thank the reviewer for pointing this out. We carried out the model-building for histones using cryo-EM maps obtained by Titan Krios. NR5A2-unbound and NR5A2-bound structures were separately determined from the same dataset by applying the three-dimensional reconstruction without symmetry. In the unbound nucleosome structure, we found the weak density map of one of the DNA ends, reflecting the DNA mobility by NR5A2-binding (**Additional fig. 1**). We further identified DNA bases from cryo-EM map density (**new Extended Figure 6**). In both structures at the same DNA region, the EM map density of DNA bases (purine and pyrimidine) near the dyad axis was completely fitted with our model (**new Extended Figure 6**). Therefore, the DNA is unambiguously assigned to the unbound nucleosome. Taken together, we conclude that NR5A2-binding slightly induces structural changes in core histones near the bound side.

Additional Fig.1 Cryo-EM map of unbound and bound nucleosome

- When examining NR5A2 binding to the native DNA sequence *in vivo*, how precise does the Mnase experiment in cells report on nucleosome position in this bulk assay? From the scale bar (Fig.4a) and it's a bit hard to see, and it seems there may be a bit of a distribution of peaks or nucleosome positions across the DNA fragment, as is common in mammalian cells. It would be nice in the current manuscript to detail the cellular experimental design and interpretation a bit more so that the reader can understand where the sequence was derived. On a related note, to identify the exact register of the native DNA sequence *in vitro*, would it be possible to take a longer fragment from this locus, ~180bp (containing the 149bp you selected as the peak nucleosome-fragment), and perform Mnase + sequencing *in vitro* +/- NR5A2? I think this experiment, if possible, could help identify more precisely the DNA register of the nucleosome *in vitro*, which may place the motif at SHL2.5, as predicted if the 149bp fragment had the energetically most favorable dyad position in the middle of the DNA sequence. Still, it could also have many different registers/translational positions on this non-601 native fragment, as described in other native sequences for CENP-B, quite some time ago: <https://doi.org/10.1074/jbc.M509666200> or more recently for OCT4: <https://doi.org/10.1038/s41586-023-06112-6>, just to list a few examples. The addition of NR5A2 and comparison of the “bound” and “unbound” registers could show if NR5A2 positions a nucleosome, like seen with other DNA-binding proteins, or, since it doesn't appear to make direct histone contacts, but instead competes with histones for DNA, it may not influence the rotational/translational nucleosome position.

We thank the reviewers for suggesting these important additional analyses.

The reviewer is correct in that there is some ambiguity with regard to the precise peak locations and nucleosome occupancy in mammalian MNase-data. We selected a region in which Nr5a2 peaks and a *SINE B1* sequence overlap as a mouse genomic locus for nucleosome reconstitution. This region was further limited to a minimal 149 bp DNA sequence based on nucleosome occupancy in mESC and 2-cell embryos. The nucleosome occupancy at this locus was depleted during the transition from an early to a late 2-cell embryo, which corresponds to the timing of major zygotic genome activation (ZGA). These results suggests that Nr5a2 promotes chromatin accessibility during ZGA at this DNA sequence (**new fig. 4a**). We provide a clarification on the detailed design of the chosen sequence in the manuscript.

As suggested by the reviewer, we reconstituted the nucleosome with 180 bp and mapped the nucleosome positioning (**new Extended Fig. 8a-c**). *In vitro* MNase assays showed that the left-position is the predominant nucleosome, in which the Nr5a2 motif is located at SHL+4 (**new Extended Figs. 8d-g and 9**). We also found that nucleosome positioning at SHL +0.5 and +2.5. NR5A2 efficiently binds to left-end positioned nucleosomes, suggesting that NR5A2 can bind to multiple positions on the nucleosomal DNA derived from a mouse genomic sequence (**new Extended Fig. 8h,i**). We further considered the interesting possibility of whether NR5A2 repositions nucleosomes, as brought up by the reviewer. We performed MNase digestion after NR5A2-binding and mapped the nucleosome positions. To exclude NR5A2-binding on linker DNA, we stringently selected the MNase-resistant fragments between 145-147 bp (**new Extended Figs. 8f,g and 9**). Our results showed that NR5A2 binding does not reposition *in vitro* reconstituted nucleosomes (**new Extended Fig. 9**). A recent pre-print suggests that histone interactions may be required for repositioning nucleosomes by pioneer factors (doi:<https://doi.org/10.1101/2023.08.25.554718>).

- While the interactions reported here with a nuclear receptor and DNA in the presence of histones are novel, I wanted to point out that minor groove competition with the histones has also been observed with SOX proteins: <https://doi.org/10.1038/s41586-020-2195-y> where “SOX11 widens the DNA minor groove.. and pulls the DNA away from the histone octamer”.

We apologize for unintentionally not including this in the original manuscript. We thank the reviewer for pointing this out and have included the minor groove competition observed by SOX11 in the discussion section.

Minor points

- In Figure 1, panel a, I would suggest to include in the SHL motif positions in the X-axis of the SeEN-seq plot, as this is the nomenclature used throughout the manuscript.

We added SHL motif position on the x-axis (**Figure 1a**).

- In the cryo-EM processing figures, it would be good to show a raw micrograph from each dataset to judge the particle distribution and a selection of 2Ds, as 2D classification was mentioned in the processing pipeline.

We added representative raw micrographs and 2D classification (**new Extended Figure 3a-d**).

- In the EXD Fig. 3 legend, the datasets are referred to as the “Krios” and “Glacios” datasets. I would also suggest including the names of the corresponding samples here. It would also be helpful to list in the legend which software was used, in addition to the Methods.

We added sample names and software in the figure legends.

- In the manuscript, it is stated that the “NR52-nucleosome complex showed that the side chain of H2A Lys75 (loop 2) was flipped and formed a new hydrogen bond with HA His82 (Fig 3a).” Is the density in the cryoEM map resolved well enough to see this side chain flip? With such a detailed analysis, it would be good to show the corresponding density in a zoomed view in both the bound and unbound complexes in the Figure.

We now included cryo-EM maps superimposed on the models (**new Figure 3a**).

- When referencing other TF-nucleosome structure papers, including OCT4 and SOX (Page 3), the BioRxiv paper has been published: <https://doi.org/10.1038/s41586-023-06112-6> and I would also include this recent paper as well: <https://doi.org/10.1038/s41586-023-06112-6>

We added these papers as references.

- In 4e, I thought it would be helpful to indicate with colors or pointers the specific regions of NR5A2, such as ZF, CTE, etc. so that one can see the CTE in this instance would also be modeled to face the histones, like in the SHL+5.5 instance.

We now show each domain with colors for clarity (**Figures 5b,c**).

Decision Letter, first revision:

Message: Our ref: NSMB-A47785A

13th Dec 2023

Dear Dr. Tachibana,

Thank you for submitting your revised manuscript "Nucleosome-bound NR5A2 structure reveals pioneer factor mechanism by minor groove anchor competition" (NSMB-A47785A). It has now been seen by the original referees and their comments are below. The reviewers find that the paper has improved in revision, and therefore we'll be happy in principle to publish it in Nature Structural & Molecular Biology, pending minor revisions to satisfy the referees' final requests and to comply with our editorial and formatting guidelines.

To facilitate our work at this stage, it is important that we have a copy of the main text as a word file. If you could please send along a word version of this file as soon as possible, we would greatly appreciate it; please make sure to copy the NSMB account (cc'ed above).

Sincerely,

Carolina Perdigoto, PhD
Chief Editor
Nature Structural & Molecular Biology
orcid.org/0000-0002-5783-7106

Reviewer #1 (Remarks to the Author):

The authors have done an outstanding job of addressing the reviewers' comments, including new experimental work. The manuscript will be an excellent contribution to the field and I recommend it for acceptance.

Reviewer #2 (Remarks to the Author):

The authors have addressed all of my comments in the revised manuscript and I highly recommend this paper for publication.

I only have one small comment regarding the MnaseI data. I think the inclusion of the MnaseI data adds a lot of value to the manuscript in understanding how transcription factors bind in dynamically positioned nucleosome environments. From my understanding, the MnaseI data show now that the most favored interaction of the complex is when the NR5A2 motif is positioned at SHL+4. The SeEN-seq profile shows only a small peak at SHL+4, so there seems to be a bit of a disconnect between the binding preferences observed in these two experiments. Yet, due to the 180bp sequence and histone-DNA interactions preferred on the native DNA sequence, it appears that the motif is constrained and is never positioned at SHL+5.5, like it is in SeEN-seq and the corresponding structure. Therefore, one cannot directly compare the binding preferences between the two experiments. Maybe it is worth explicitly noting that in the native sequence, NR5A2 is not presented with a motif at SHL+5.5, but if it were, the prediction would be that it would also be a preferred binding site. The other two positions that also seem to be available to be bound in the native sequence, SHL+0.5 and SHL+2.5 align with peaks in the SeEN-seq, so you could also mention this.

Author Rebuttal, first revision:

We would like to sincerely thank all reviewers for supporting our revised manuscript and the constructive suggestions throughout the review process. For clarity, we have highlighted the new text addressing the remaining comments in yellow in the manuscript.

Please find our detailed responses below for minor revisions.

Reviewer #1:**Remarks to the Author:**

The authors have done an outstanding job of addressing the reviewers' comments, including new experimental work. The manuscript will be an excellent contribution to the field and I recommend it for acceptance.

We are very grateful to the reviewer for supporting our work. Thank you very much.

Reviewer #2:**Remarks to the Author:**

The authors have addressed all of my comments in the revised manuscript and I highly recommend this paper for publication.

I only have one small comment regarding the MnaseI data. I think the inclusion of the MnaseI data adds a lot of value to the manuscript in understanding how transcription factors bind in dynamically positioned nucleosome environments. From my understanding, the MnaseI data show now that the most favored interaction of the complex is when the NR5A2 motif is positioned at SHL+4. The SeEN-seq profile shows only a small peak at SHL+4, so there seems to be a bit of a disconnect between the binding preferences observed in these two experiments. Yet, due to the 180bp sequence and histone-DNA interactions preferred on the native

DNA sequence, it appears that the motif is constrained and is never positioned at SHL+5.5, like it is in SeEN-seq and the corresponding structure. Therefore, one cannot directly compare the binding preferences between the two experiments. Maybe it is worth explicitly noting that in the native sequence, NR5A2 is not presented with a motif at SHL+5.5, but if it were, the prediction would be that it would also be a preferred binding site. The other two positions that also seem to be available to be bound in the native sequence, SHL+0.5 and SHL+2.5 align with peaks in the SeEN-seq, so you could also mention this.

We sincerely thank the reviewer for supporting our work. Based on your excellent suggestion, we have added one paragraph discussing the different results between 601-based SeEN-seq and the native sequence. Further structural analysis focusing on how NR5A2 engages with the endogenous nucleosome at SHL+0.5, +2.5 and +4 would be interesting in the future.

Final Decision Letter:

Message 31st Jan 2024

:
Dear Dr. Tachibana,

We are now happy to accept your revised paper "Nucleosome-bound NR5A2 structure reveals pioneer factor mechanism by minor groove anchor competition" for publication as a Article in Nature Structural & Molecular Biology.

To assist our authors in disseminating their research to the broader community, our SharedIt initiative provides all co-authors with the ability to generate a unique shareable

link that will allow anyone (with or without a subscription) to read the published article. Recipients of the link with a subscription will also be able to download and print the PDF.

Your paper will be published online soon after we receive proof corrections and will appear in print in the next available issue. You can find out your date of online publication by contacting the production team shortly after sending your proof corrections.

Please note that *Nature Structural & Molecular Biology* is a Transformative Journal (TJ). Authors may publish their research with us through the traditional subscription access route or make their paper immediately open access through payment of an article-processing charge (APC). Authors will not be required to make a final decision about access to their article until it has been accepted. http://www.nature.com/protocolexchange/about

<https://www.springernature.com/gp/open-research/transformative-journals>> Find out more about Transformative Journals

Authors may need to take specific actions to achieve compliance with funder and institutional open access mandates. If your research is supported by a funder that requires immediate open access (e.g. according to Plan S principles) then you should select the gold OA route, and we will direct you to the compliant route where possible. For authors selecting the subscription publication route, the journal's standard licensing terms will need to be accepted, including self-archiving policies. Those licensing terms will supersede any other terms that the author or any third party may assert apply to any version of the manuscript.

Sincerely,

Carolina Perdigoto, PhD
Chief Editor
Nature Structural & Molecular Biology
orcid.org/0000-0002-5783-7106